# Distinct local and global functions of mouse Aβ low-threshold mechanoreceptors in mechanical nociception

Mayank Gautam [1], Akihiro Yamada[2,3], Ayaka I. Yamada[2,3], Qinxue Wu[1], Kim Kridsada[1], Jennifer Ling[2], Huasheng Yu[1], Peter Dong[1], Minghong Ma [1], Jianguo Gu [2] ✉ & Wenqin Luo [1] ✉

The roles of Aβ low-threshold mechanoreceptors (LTMRs) in transmitting mechanical hyperalgesia and in alleviating chronic pain have been of great interest but remain contentious. Here we utilized intersectional genetic tools, optogenetics, and high-speed imaging to specifically examine functions of *Split^Cre* labeled mouse Aβ-LTMRs in this regard. Genetic ablation of Split^Cre-Aβ-LTMRs increased mechanical nociception but not thermosensation in both acute and chronic inflammatory pain conditions, indicating a modality-specific role in gating mechanical nociception. Local optogenetic activation of Split^Cre-Aβ-LTMRs triggered nociception after tissue inflammation, whereas their broad activation at the dorsal column still alleviated mechanical hypersensitivity of chronic inflammation. Taking all data into consideration, we propose a model, in which Aβ-LTMRs play distinctive local and global roles in transmitting or alleviating mechanical hyperalgesia of chronic pain, respectively. Our model suggests a strategy of global activation plus local inhibition of Aβ-LTMRs for treating mechanical hyperalgesia.

Chronic pain is a devastating disorder of the nervous system, affecting more than 30% people worldwide[1]. A prominent symptom of chronic pain is mechanical hyperalgesia[2,3], or increased pain triggered by mechanical stimuli. At present, effective treatment for mechanical hyperalgesia is limited, reflecting a knowledge gap in its underlying mechanisms.

Aβ low-threshold mechanoreceptors (Aβ-LTMRs) are large-diameter, highly-myelinated, and fast-conducting primary somatosensory neurons, which normally mediate tactile, discriminative touch, and vibration sensation. Aβ-LTMRs are further divided into rapidly adapting (RA) and slowly adapting (SA) types based on their firing patterns in response to a sustained mechanical stimulus. At baseline conditions, co-activation of touch- and pain-sensing neurons inhibits the flow of nociceptive information[4], as proposed in the "gate control theory of pain"[5].

Functions of Aβ-LTMRs in chronic pain have been of great interest but are controversial. On one hand, several studies suggested that Aβ-LTMRs may mediate mechanical hyperalgesia of chronic pain. Spinal nerve ligation elicited increased irregular firings of rat RA Aβ-LTMRs[6,7]. When activities of Aβ-LTMRs were masked using nerve block, mechanical hyperalgesia was abolished in patients or mice with nerve injury[8,9]. Moreover, spinal nerve ligation-induced mechanical allodynia in rats was abolished after destroying the dorsal column, where ascending axons of Aβ-LTMRs project through[10]. On the other hand, some studies suggested that Aβ-LTMRs inhibit nociception even in chronic conditions. Rubbing or massaging of a painful area of the skin, which presumably activates Aβ-LTMRs, attenuated pain in humans[11]. The spinal cord stimulator (SCS) and transcutaneous electrical nerve stimulation (TENS), which was developed based on the "gate control theory of pain"[5] to target Aβ-LTMRs[12], are effective for treating various

[1]Department of Neuroscience, Perelman School of Medicine, University of Pennsylvania, Philadelphia, PA 19104, USA. [2]Department of Anesthesiology and Perioperative Medicine, School of Medicine, University of Alabama at Birmingham, Birmingham, AL 35294, USA. [3]These authors contributed equally: Akihiro Yamada, Ayaka I. Yamada. ✉e-mail: jianguogu@uabmc.edu; luow@pennmedicine.upenn.edu

chronic pain conditions, including neuropathic pain[13]. Additional real-life therapy procedures, including massage therapy[14] and electroacupuncture[15], presumably involve the activation of Aβ-LTMRs for their beneficial effects.

When comparing different studies and their results, we noticed two factors, which likely contribute to the diverse outcomes of Aβ-LTMRs in transmitting mechanical hyperalgesia or in its alleviation. The first consideration is the specificity of Aβ-LTMR manipulation and the functional readouts. Though pharmacological and mechanical/electrical stimuli can inhibit or activate Aβ-LTMRs, other types of nerve fibers are likely simultaneously affected as well. For genetic and optogenetic manipulations, few available mouse genetic tools are selective or show high preference for Aβ-LTMRs. In addition, it has been challenging to differentiate touch and nociception associated reflex behavioral responses in animal studies. The second consideration is whether Aβ-LTMRs are manipulated locally at the area affected by chronic pain or in a broad manner including the unaffected Aβ-LTMRs, which may generate different sensory and behavioral outcomes.

To provide novel insight into this question, we utilized tools with improved specificity, including intersectional mouse genetics, opto-tagged electrophysiological recordings, and optogenetic activation of Aβ-LTMRs, and we performed a battery of behavioral tests to probe mouse mechanical and thermal sensations when Aβ-LTMRs were manipulated locally or globally. We also took advantage of high-speed imaging method that our lab previously established[16] to differentiate touch-related non-nociceptive behavior from nocifensive behaviors. Here, we found that mice with global ablation of Split$^{Cre}$-Aβ-LTMRs showed decreased sensitivity to gentle mechanical forces, increased mechanical nociception at baseline condition, and increased mechanical hyperalgesia in a chronic inflammatory pain model, suggesting that Aβ-LTMRs function to inhibit mechanical nociception and mechanical hyperalgesia. Thermosensation was not affected by Aβ-LTMRs manipulation, indicating that these afferents functioned in a modality-specific manner. In reverse experiments, local optical activation of Split$^{Cre}$-Aβ-LTMRs at the hind paw triggered nocifensive responses in chronic inflammatory and neuropathic pain models. This suggests that activities of locally affected Aβ-LTMRs lead to nociception, likely contributing to mechanical hyperalgesia of chronic nociception. Finally, global activation of Aβ-LTMRs at the dorsal column alleviated mechanical hyperalgesia in a chronic inflammatory pain model. Together, our results establish a model that Aβ-LTMRs play globally inhibitory but locally promoting roles for mechanical hyperalgesia. Our model suggests a strategy, global activation plus local inhibition of Aβ-LTMRs, for treating mechanical hyperalgesia of chronic pain.

## Results

### Histological and electrophysiological characterizations of Split$^{Cre}$-Aβ ReChR mice confirmed that Split$^{Cre}$ preferentially recombined in Aβ-LTMRs

To clarify functions of Aβ-LTMRs in mechanical hyperalgesia, we first established a mouse genetic strategy that would allow specific manipulation of Aβ-LTMRs. Previous studies, including our own, suggests that Split$^{Cre}$ likely recombines in Aβ-LTMRs[17,18]. The Split$^{Cre}$ mouse line was generated by GENSAT[19], using two bacterial artificial chromosomes (BACs) containing the *Abhd3* and *Ntng2* genes. The two halves of iCre (19-59 and 60-343) were fused with the constitutively active coiled-coil interaction domain of the yeast transcription factor GCN4 and were inserted at the start codon of the *Abhd3* and *Ntng2* genes. With this design, the iCre recombines in cells where both *Abhd3* and *Ntng2* transgenes are expressed (Supplementary Fig. 1A).

We have also noticed that Split$^{Cre}$ recombined in a small number of non-neuronal glia cells. Thus, we utilized an intersectional genetic strategy (Supplementary Fig. 1B) by breeding together Split$^{Cre\ 17}$ and

*Advil$^{FlpO\ 20}$* alleles, in which the Flippase (Flp) expression was restricted to peripheral ganglion neurons. The double heterozygous mice were then crossed with a Cre and Flp double-dependent Red-activatable Channelrhodopsin (ReChR) reporter allele (*Rosa$^{ReChRf/f}$*)[21] to generate triple heterozygous Split$^{Cre}$;*Advil$^{FlpO}$;Rosa$^{ReChRf/+}$* (Split$^{Cre}$-Aβ ReChR) mice. RNAscope with dorsal root ganglion (DRG) sections of the triple mice confirmed that almost all (96%) Eyfp$^+$ (tagged to *ReChR*) neurons co-expressed *iCre* (Supplementary Fig. 1C, D). Since *iCre* cDNA were split into N- (120 bp) and C-fragments (849 bp), and since the C-fragment contained the majority of cDNA, C- but not N-fragments of *iCre* were visualized by RNAscope in situ hybridization. Around half of DRG neurons (48.9%) expressing the *iCre* C-fragment were positive for *Eyfp* (Supplementary Fig. 1D), demonstrating the intersectional effect of this split Cre strategy. In addition, immunohistochemistry with DRG sections revealed that expression of EYFP (recognized by anti-GFP antibodies) rarely overlapped with nociceptor markers CGRP (5.44% of GFP$^+$ DRG neurons) and IB4 (4.05%) (Supplementary Fig. 1E, I) but showed high overlap with a myelinated neuron marker NF200 (89.32%) and an Aβ-LTMR marker RET (74.35%)[22,23]. There was also some overlap of expression with the proprioceptor marker PV (12.9%) (Supplementary Fig. 1F–I). Moreover, we performed multiplex RNAscope in situ hybridization (Fig. 1A–D) using additional markers for Aβ-RA LTMRs, *Ret* and *Calbindin*, or Aδ/Aβ LTMRs, *Ntrk2*[24,25]. Around 85% of Eyfp+ DRG neurons were Ret$^+$, ~67% Eyfp+ DRG neurons were Calb1+, and ~40% of Eyfp+ neurons co-expressed *Ntrk2*. Together, molecular characterization of genetically labeled DRG neurons suggests that Split$^{Cre}$ recombines in large-diameter DRG neurons, mainly Aβ-RA LTMRs.

Consistently, immunostaining of lumbar spinal cord sections showed that only few GFP$^+$ central terminals innervate layers I/II, labeled by CGRP and IB4 staining (Fig. 1E), but they were enriched in deeper dorsal horn laminae (III-V), indicated by VGLUT1$^+$ staining (Fig. 1F). Immunostaining of plantar skin sections showed that most GFP$^+$ peripheral axons were NF200$^+$ (Fig. 1G and Supplementary Fig. 1J), and they innervated a large percentage of S100$^+$ Meissner's corpuscles (a type of RA Aβ-LTMRs, Fig. 1H and Supplementary Fig. 1K, V, W). Few (7.7%) GFP$^+$ axons innervate K8$^+$ Merkel cells (Merkel cell-neurite complex), a type of SA Aβ-LTMRs, Fig. 1I and Supplementary Fig. 1L, V, W). CGRP$^+$ fibers also innervate dermal papillae skin regions, but GFP+ and CGRP+ fibers were non-overlapping (Supplementary Fig. 1M). Whole-mount immunostaining of hairy skin revealed that GFP$^+$ peripheral axons are NF200$^+$ (Supplementary Fig. 1N). A few GFP$^+$ fibers innervate Merkel cells in touch domes of the hairy skin (Supplementary Fig. 1O–Q), and ~11% of touch domes in the back hairy skin were innervated by GFP$^+$ fibers (Supplementary Fig. 1W). Many GFP$^+$ fibers formed lanceolate ending structures around hair follicles (RA Aβ-LTMRs, Supplementary Fig. 1R–T). Taken together, the histological data suggest that this intersectional genetic strategy induce ReChR expression preferentially in Aβ-LTMRs. In addition, RA Aβ-LTMRs were prominently labeled in the glabrous skin, but RA and some SA Aβ-LTMRs were labeled in the hairy skin of these mice.

To reveal electrophysiological properties of ReChR-labeled sensory afferents, we conducted opto-tagged electrophysiological recordings. We first performed patch-clamp recordings at the Node of Ranvier of ReChR/YFP$^+$ fiber in isolated saphenous nerves (Fig. 2A). Action potentials were evoked by electrical stimulation at the distal site of the nerves (Fig. 2B), and conduction velocities were determined. The conduction velocity of randomly recorded ReChR/YFP$^+$ fibers was ~20 m/s ($n = 17$), in the range of Aβ fibers (Fig. 2C). Next, we used an ex-vivo skin-nerve preparation with both hind paw glabrous skin and paw hairy skin and performed single-unit recordings to determine mechanical threshold and firing patterns of ReChR/YFP$^+$ fibers. ReChR/YFP$^+$ units were first identified by their responses to orange (605 nm) LED light stimuli. Mechanical stimuli by an indentator were then applied to the same receptive field as light stimuli (Fig. 2D, E). A

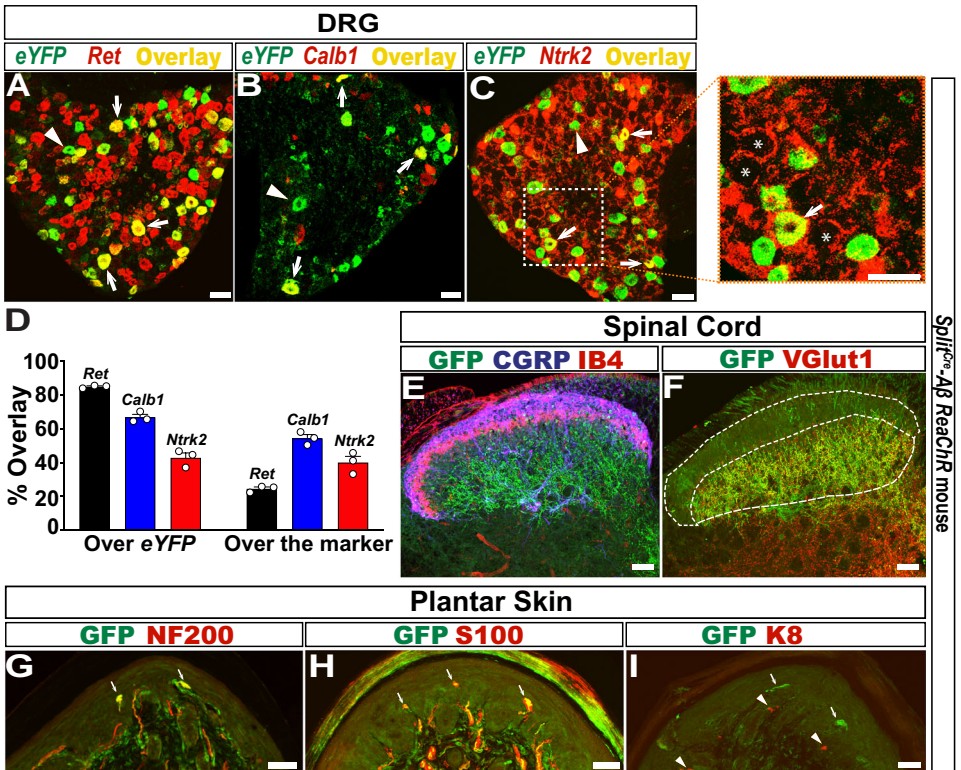

**Fig. 1 | Histological characterization of *Split^Cre^-ReaChR* mice. A–C** RNAscope of *Eyfp* (ReaChR) with *Ret, Calb,* or *Ntrk2* using DRG section of *Split^Cre^-ReaChR* mice. White arrows and arrow heads indicate some overlapped and non-overlapped *Eyfp^+^* neurons, respectively. The zoomed image of (**C**) shows high expression of *Ntrk2* in some DRG neurons and satellite glial cells surrounding Ntrk2 negative neurons (marked by asterisk). Scale bar = 50 μm in all micrographs until otherwise mentioned. **D** Quantification of the percentage of *Eyfp^+^* neurons expressing different markers and vice versa. 4–5 sections/mouse, *n* = 3 mice. Error bars represent Mean ± S.E.M. **E** Triple immunostaining of a lumbar spinal cord section showing laminar segregation of GFP^+^ central terminals and CGRP^+^ or IB4^+^ nociceptor terminals. **F** Double immunostaining of lumbar spinal cord section showing GFP^+^ central terminals innervate deeper dorsal horn lamina (III-V), indicated by VGlut1^+^ staining. **G–I** Double immunostaining of plantar skin sections showing that GFP^+^ peripheral axons are NF200^+^ and innervate Meissner's corpuscles (revealed by S100 staining) but not K8^+^ Merkel cells. White arrows indicate some Meissner's corpuscles, while white arrowheads indicate Merkel cells. 6–8 sections/mouse, *n* = 4–6 mice. Source data are provided as a Source Data file.

total of 7 light-sensitive saphenous nerve recordings and 9 light-sensitive tibia nerve recordings were recorded from the hairy and the glabrous skin, respectively. 4 units were RA (Fig. 2F), and 3 units were SA (Fig. 2G) in the hairy skin, while 8 units were RA, and 1 unit was SA (Fig. 2H) in the glabrous skin. Mechanical stimulation-response curves of ReaChR^+^ units showed typical RA and SA firing patterns (Fig. 2I). The mechanical thresholds of light sensitive units in the hairy and glabrous skin were similar, 1.4 ± 0.5 mN (*n* = 11) from RA units and 3.9 ± 1.0 (*n* = 4) from SA units (Fig. 2J), which were within the threshold of LTMRs. Taken all into consideration, the molecular, histological, and electrophysiological characterizations reveal that this intersectional genetic strategy specifically labels cutaneous Aβ-LTMRs in both glabrous and hairy skin and that most of them are RA Aβ-LTMRs in the hind paw glabrous skin.

## Ablation of Aβ-LTMRs by DTA treatment of *Split^Cre^-Aβ TauDTR* mice

Next, we crossed *Split^Cre^;Advil^FlpO^* double mice to homozygous reporter mice, which contained a Cre- and FlpO- double-dependent DTR allele (*Tauds-DTR^f/f^*) (the human diphtheria toxin receptor (DTR) driven by a pan neuronal Tau promoter)[26] and a Cre-dependent Ai9 (*Rosa-tdTomato^f/f^*) reporter allele. The resulting quadruple mice expressed DTR in neurons co-expressing *Split^Cre^* and *Advil^FlpO^*, while tdTomato expression indicated the recombination activity of *Split^Cre^*. The quadruple progenies were regarded as *Split^Cre^-Aβ TauDTR* mice (Fig. 3A). Similar to ReaChR2, the expression of tdTomato highly overlapped

with large diameter neuronal marker NF200 (86.7% of tdTomato^+^ neurons), to a lesser extent with PV (12.13%), but rarely overlapped with nociceptive markers CGRP (4.01%) and IB4 (3.03%) (Supplementary Fig. 2A–D). Immunostaining of lumbar spinal cord section showed that tdTomato^+^ central terminals mainly innervate deep (VGLUT1^+^) but not superficial (CGRP^+^ and IB4^+^) dorsal horn laminae (Supplementary Fig. 2E, F). tdTomato^+^ fibers also innervated RA and SA Aβ-LTMRs in the glabrous and hairy skin (Supplementary Fig. 2G–N). Together, these histological results validated the preferential recombination in Aβ-LTMRs of *Split^Cre^-Aβ TauDTR* mice.

To ablate Split^Cre^-Aβ-LTMRs, six-week-old *Split^Cre^-Aβ TauDTR* mice were intraperitoneally injected with either DTA or water for injection (vehicle control) daily for 1 week. Two weeks after the last injection, mice were sacrificed to confirm the ablation efficiency. The numbers of *tdTomato^+^* neurons per lumbar DRG sections (Fig. 3B, C) were quantified, and DTA treatment led to a ~70% ablation compared to the vehicle group (*p* < 0.0001) (Fig. 3D). Specifically, significant reductions were found for tdTomato+ neurons co-expressing *Ret* and *Calb* but not *Ntrk2* (Fig. 3K–O). In addition, the mean fluorescence intensity of tdTomato^+^ central terminals in the deeper dorsal horn of the lumbar spinal cord of DTA-treated mice decreased 63% (*p* < 0.0001) (Fig. 3E–G). Moreover, though Meissner corpuscles were not ablated by this treatment (Supplementary Fig. 2O, P, S), the percentage of Meissner corpuscles innervated by tdTomato+ fibers (Fig. 3H–J) and the total number of tdTomato^+^ Meissner's corpuscles-like structures in dermal papilla (Supplementary Fig. 2Q, R, T) were significantly

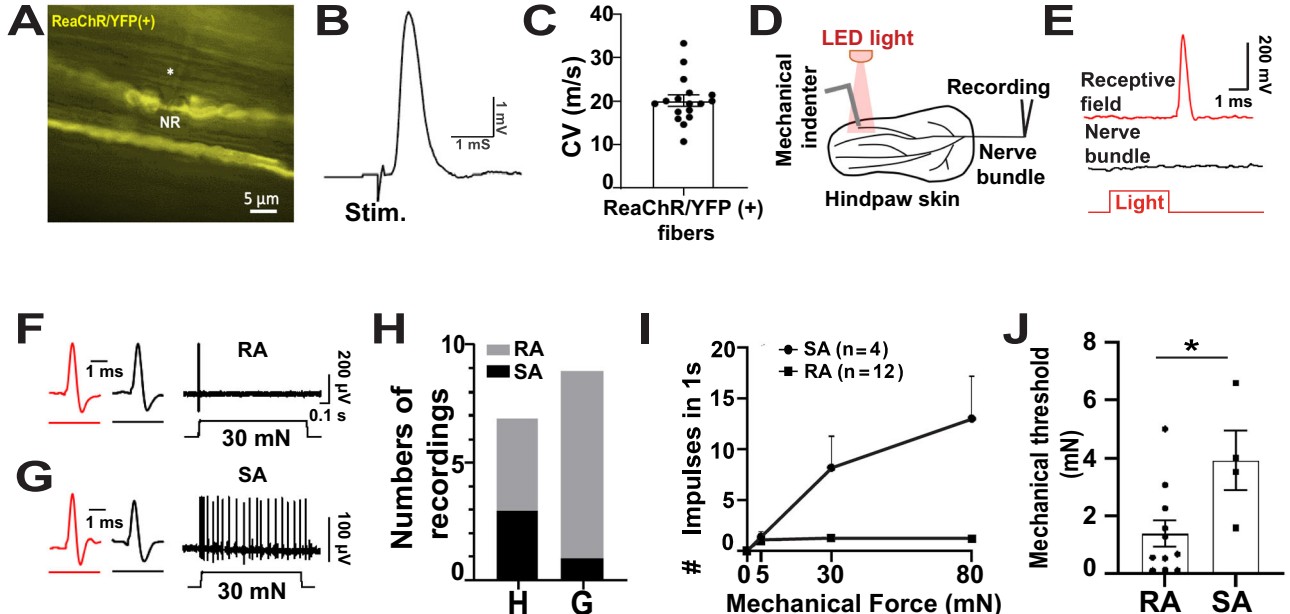

**Fig. 2 | Electrophysiological characterization of *Split^Cre*-ReaChR mice. A** Image showing two ReaChR/YFP⁺ fibers in an ex-vivo saphenous nerve preparation, one fiber had a node of Ranvier (NR) where patch-clamp recording was made to record action potential (AP) impulses. The asterisk indicates the shadow of the recording electrode tip. Scale bar = 5 μm. **B** Sample trace showing AP impulse recorded from a ReaChR/YFP⁺ fiber. The AP impulse was evoked by an electrical stimulus at a distal site of the saphenous nerve. A stimulation artifact (Stim.) is shown prior to the AP impulse. **C** Summary data of conduction velocities of AP impulses in ReaChR/YFP⁺ fibers propagated from the stimulation sites to the recording sites ($n = 17$ fibers). **D** Diagram illustrating pressure-clamped single-fiber recordings of AP impulses evoked by LED light stimulation and mechanic indentation at the same receptive field in an ex-vivo hind paw skin-nerve preparation. **E** Top panel, sample trace showing an AP impulse evoked by LED light in a receptive field of the hairy skin of the skin-nerve preparation. Middle panel, no AP impulse could be evoked by light stimulation at saphenous nerve bundle. Bottom panel, LED light pulse. The representative traces showing rapidly adapting (RA, (**F**)) and slowly adaptive (SA, (**G**)) AP impulses evoked by mechanical indentation (30 mN) at two different receptive fields where LED lights also evoked AP impulses (red traces at an expanded time scale). The black traces in **F** and **G** after the red traces are mechanically evoked AP impulses at an expanded time scale. Glabrous skin-tibia nerve preparation was used in (**F**), hairy skin-saphenous nerve preparation was used in (**G**). **H** Numbers of light-sensitive RA (Grey bar) and light-sensitive SA units (Black bar) recorded from the hairy/H ($n = 7$) and the glabrous skin/G ($n = 9$). **I** Mechanical stimulation-response curves of light-sensitive RA and light-sensitive SA units. SA; $n = 4$ except $n = 3$ at 80 mN. RA; $n = 12$. **J** Comparison of mechanical thresholds of light sensitive RA ($n = 11$) and SA units ($n = 4$, $p = 0.019$). Data represent Mean ± S.E.M., *$p < 0.05$, two-sided Student's $t$ tests. Source data are provided as a Source Data file.

reduced. Thus, efficient ablation of Split^Cre-Aβ-LTMRs were achieved using this genetic and pharmacological strategy.

## Ablation of Split^Cre-Aβ-LTMRs reduced gentle touch sensation but increased mechanical nociception in the glabrous skin

To investigate the requirement of Aβ-LTMRs in acute and chronic nociceptive sensations, we conducted a battery of mouse behavior assays with control and ablated mice. We first examined general locomotor behavior of mice with the open field test (Supplementary Fig. 3A). Ablation of Split^Cre-Aβ-LTMRs did not significantly alter the time spent in peripheral (vehicle = 1098 ± 16.7 s vs. DTA = 1075 ± 15.7 s, $p = 0.38$) and central zones (vehicle = 103 ± 16.8 s vs. DTA = 125.3 ± 15.8 s, $p = 0.328$). There was also no significant difference between the two groups in the total distance travelled (vehicle = 62.32 ± 3.1 m vs. DTA = 64.05 ± 4.0 m, $p = 0.72$). These results suggest that ablation of Split^Cre-Aβ-LTMRs do not significantly change mouse general locomotor functions or generate obvious anxiety-associated behaviors.

Next, we tested mechanosensitivity of mouse hind paws of control and ablated mice (Fig. 4A). Ablation of Split^Cre-Aβ-LTMRs did not alter 50% paw-withdrawal mechanical threshold (PWT, static mechanosensitivity, vehicle = 0.99 ± 0.005 g, DTA = 0.98 ± 0.007 g, $p > 0.99$, $n = 9$ mice) (Fig. 4B) measured by the von Frey hair (VFH) test. A likely reason for no change of 50% PWT in the ablated mice is that other types of mechanosensory afferents in the hind paw, such as SA Aβ-LTMRs and MrgprD⁺ afferents, are largely spared by this genetic strategy. On the other hand, sensitivity to dynamic gentle touch, measured as the percentage paw withdrawal in response to by a

dynamic cotton swab (Fig. 4C), was significantly attenuated in the ablated mice (vehicle = 91.11 ± 4.84% vs. DTA = 51.11 ± 8.88%, $p = 0.0014$, $n = 9$ mice). The ablated mice also spent a significantly longer time attempting to remove a sticky tape attached to the paw, which generated small amounts of mechanical forces (vehicle = 87.6 ± 14.33 s vs. DTA = 198.1 ± 32.74 s, $p = 0.0063$, $n = 10$ mice) (Fig. 4D and supplementary movie 1). Together, these results suggest that the ablated mice are less sensitive to dynamic gentle mechanical forces. Moreover, when mice were tested using a chamber with different floor textures, hook (rough) vs. loop (smooth) (Supplementary Fig. 3E), the texture preference was altered in the ablated mice. Compared to control mice, which showed no significant preference for the floor textures (Hook = 386.5 ± 23.38 vs. Loop = 478.4 ± 56.54 seconds, $p = 0.505$), the ablated mice preferred to stay in the smoother loop surface compartment (Hook = 323.9 ± 36.95 vs. Loop = 559.9 ± 26.59 seconds, $p = 0.001$, $n = 8$ (vehicle) and 7 (DTA) mice) (Fig. 4E). In short, the behavioral deficits in dynamic gentle mechanical forces and tactile preference of DTA-treated mice supported significant functional disruption of Aβ-LTMRs in these mice.

Next, we tested mechanical nociception at the paw. In response to an alligator clip at the plantar skin, licking episodes, which reflect mouse nociception, significantly increased in the ablated mice (vehicle = 4.6 ± 1.01 episodes/1 min. vs. DTA = 13.8 ± 1.98 episodes/1 min., $p = 0.0002$, $n = 10$ mice in each group, Fig. 4F and supplementary movie 2). This result suggests that Aβ-LTMRs function to inhibit mechanical nociception and that disruption of Aβ-LTMR functions results in mechanical hyperalgesia. This might also

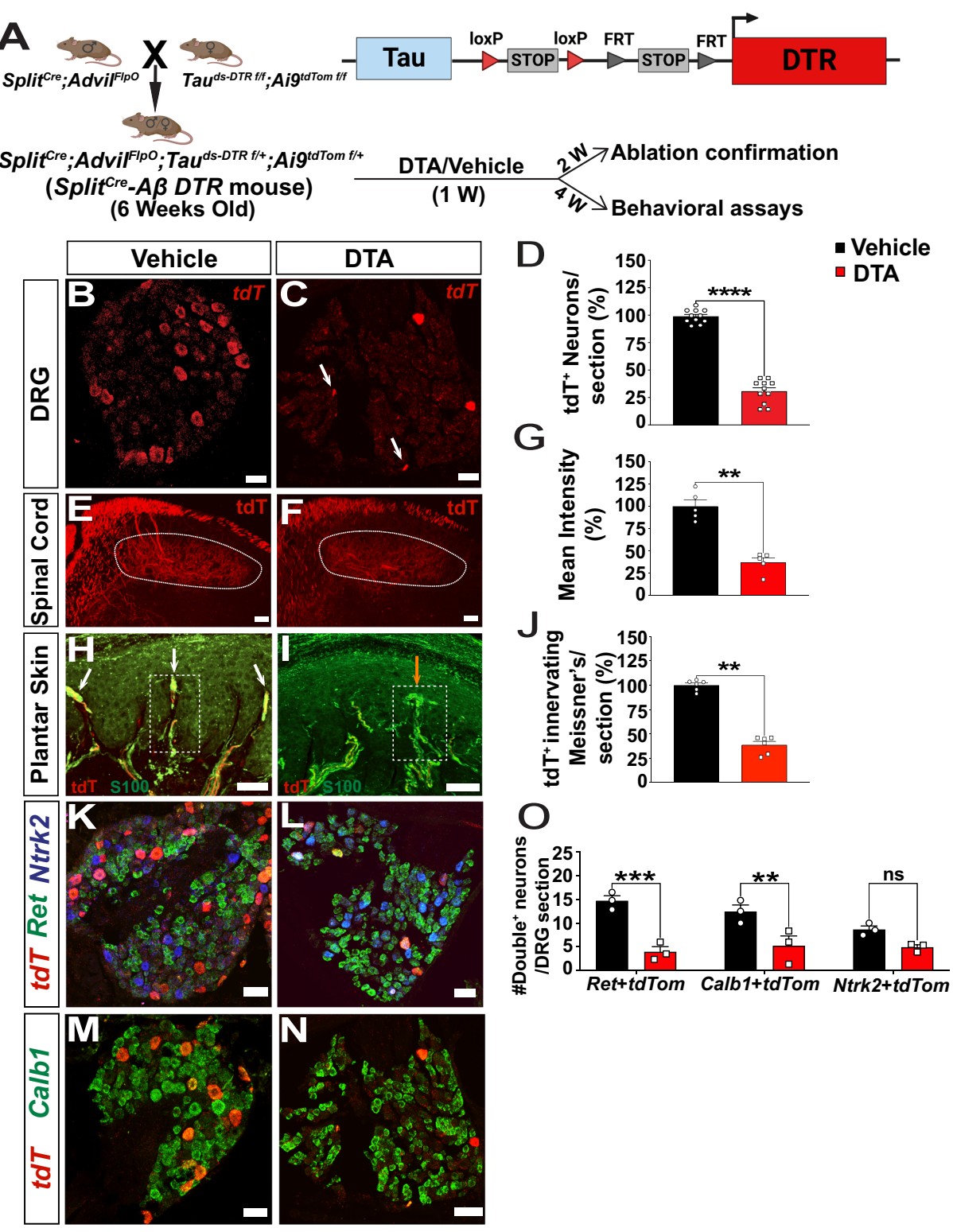

explain why the ablated mice tended to avoid the rough surface in the floor texture assay. In contrast, no difference in thermal nociception, tested by static hot plate (Supplementary Fig. 3B), dynamic hot plate (Supplementary Fig. 3C), and dry ice test (Supplementary Fig. 3D) (see also supplementary table 1), was found between the two groups, suggesting that Aβ-LTMRs inhibit nociception in a modality-specific manner.

Moreover, we tested mechanical and heat hyperalgesia in a CFA-induced inflammatory pain model. In contrast to the baseline condition, the ablated mice showed significant reductions in 50% PWT (Fig. 4G), indicating an increased mechanical nociception. The PWT significantly decreased in DTA-treated mice at 2 h (vehicle = $0.54 \pm 0.12$ g vs. DTA = $0.05 \pm 0.14$ g, $p = 0.0149$), 14th day (vehicle = $0.18 \pm 0.06$ g vs. DTA = $0.037 \pm 0.007$ g, $p = 0.0427$), 21st day (vehicle =

**Fig. 3 | Pharmacogenetic ablation of Split^Cre-Aβ-LTMRs. A** Illustration showing the intersectional genetic strategy to generate *Split^Cre-Aβ TauDTR* mice and the protocol timeline. RNAscope of *tdTomato* with lumbar DRG sections showing *tdTomato+* DRG neurons in vehicle (**B**) and DTA-treated (**C**) mice. **D** Quantification of the *tdTomato+* neurons per DRG showed a significant decrease with DTA treatment, *p* < 0.0001. The data were normalized to the average number of *tdTomato+* DRG neurons/section of control mice. 3–4 sections/mouse, *n* = 3 mice in both groups. **E, F** Images showing tdTomato+ central terminals in the dorsal horn of the lumbar spinal cord of vehicle or DTA treated mice. Dotted white area indicates laminae III-V. **G** Quantification of the mean tdTomato+ fluorescence intensity in laminae III-V showed a significant decrease with DTA treatment, *p* = 0.0079. The data were normalized to the average mean intensity of Tdt+ central terminals of control mice. 6-8 sections per mouse, *n* = 5 mice. **H, I** Images showing tdTomato+ innervation in the Meissner's corpuscles of vehicle or DTA-treated mice. White arrowheads indicate Meissner's corpuscles innervated with tdTom+ fibers, and red arrow points to the Meissner's corpuscle without tdTomato+. **J** Quantification of the number of tdTomato+ fibers innervating Meissner's corpuscles per footpad section showed a significant decrease with DTA treatment, *p* = 0.0022. The data were normalized to the average number of tdTomato+ fibers innervating dermal papillae of control mice. 5–6 sections per mouse, *n* = 6 mice. **K, L** RNAscope of *tdTomato*, *Ret*, and *Ntrk2+* with lumbar DRG sections of vehicle or DTA-treated mice. **M, N** RNAscope of *tdTomato* and *Calb1* with the lumbar DRG sections of vehicle or DTA-treated mice. **O** Quantification of the double positive neurons per DRG section showed a significant decrease in those co-expressing *Ret* (*p* = 0.0002) and *Calb1* (*p* = 0.004) but not *Ntrk2* (*p* = 0.16). 4 sections per mouse, *n* = 3 mice. Scale bar represents 50 μm in all micrographs. Error bars represent Mean ± S.E.M. Unpaired, two-tailed Mann-Whitney test. **p < 0.01, ****p < 0.0001, ns = non-significant. Source data are provided as a Source Data file.

0.38 ± 0.12 g vs. DTA = 0.049 ± 0.015 g, *p* = 0.0012), and 28th day (vehicle = 0.694 ± 0.086 g vs. DTA = 0.052 ± 0.018 g, *p* = 0.0012). The differences at days 1, 3, and 7 were not significant due to the floor effect (the PWTs of control mice were already close to 0). Interestingly, the ablated mice showed no significant change in thermal hyperalgesia (Fig. 4H), which further supported the modality-specific function of Aβ-LTMRs. We also tested mechanical hyperalgesia in a neuropathic Medial Plantar Nerve Ligation (MPNL) model. However, since mechanical thresholds of the control mice were close to 0 from 3 days after the nerve ligation and didn't recover during the experimental period (floor effect), no further reduction was observed in the ablated group (Supplementary Fig. 3F). Together, these behavior results suggest that in the glabrous skin, disruption of Split^Cre-Aβ-LTMRs reduces the dynamic gentle-touch sensitivity and specifically dis-inhibits mechanical nociception at baseline and chronic inflammatory conditions.

## Ablation of Split^Cre-Aβ-LTMRs altered gentle touch sensation and mechanical nociception in the hairy skin

The hairy skin contains Aβ-LTMRs as well as C- and Aδ- LTMRs[27] (Supplementary Fig. 4A). Thus, behavior outcomes from the hairy skin of ablated mice could be more complicated than the glabrous skin. We conducted a similar sticky tape test, which generated gentle mechanical stimuli at the de-haired back skin. The ablated mice displayed significantly increased scratching behavior in response to the tape (vehicle = 0.181 ± 0.181 scratch bouts/5 min vs. DTA = 11 ± 3.964 scratch bouts/5 min, *p* = 0.0027, *n* = 11 and 10 mice in vehicle and DTA groups, respectively) (Supplementary Fig. 4B), but no difference in back-attending episodes between the vehicle and ablated groups (vehicle = 17.64 ± 3.8 attending episodes/5 min vs. DTA = 19.2 ± 3.552 attending episodes/5 min, *p* = 0.743) was observed (Supplementary Fig. 4C). These results suggest that the ablated mice show increased responses to gentle mechanical forces at the hairy skin, which might be mediated by dis-inhibited C- and/or Aδ-LTMRs. In addition, in response to the application of an alligator clip at the neck nape, the ablated mice displayed significantly increased number of attending episodes (vehicle = 8.45 ± 1.60 vs. DTA = 17.4 ± 2.16 attending episodes/1 min, *p* = 0.0074, *n* = 11 and 10 mice in vehicle and DTA groups, respectively) (Supplementary Fig. 4D and supplementary movie 3), indicating an increased mechanical nociception. No difference in heat nociception between the groups was found, tested by tail-immersion assay at different temperatures (supplementary table 1). These data indicate that Split^Cre-Aβ-LTMRs are required to "gate" both gentle touch transmission and mechanical nociception in the hairy skin.

## Local optogenetic activation of Split^Cre-Aβ-LTMRs evoked nocifensive behaviors in chronic inflammatory and neuropathic pain models

Our results using the ablated mice revealed functions of Aβ-LTMRs in specifically inhibiting mechanical nociception. Since multiple studies also suggested functions of Aβ-LTMRs in transmitting mechanical hyperalgesia[8,28–30], we conducted reverse experiments using optogenetic activation of Aβ-LTMRs. Peripheral optogenetic stimulation of *Split^Cre-Aβ ReChR* mice (Fig. 5A) would activate a small population of Aβ-LTMRs innervating the skin area targeted by the blue laser light. We first tested different intensities (5, 10, and 20 mW) of blue laser (473 nm) over several skin areas, the paw, nape of the neck, back and tail, to characterize spontaneous laser-evoked behaviors using a high-speed camera[16,31] (Supplementary Fig. 5A). *Split^Cre-Aβ ReChR* mice were very responsive to peripheral laser stimuli. They showed ~40-90% hind paw withdrawal responses (Fig. 5B), and ≥80% responses at hairy skin with different intensities (Supplementary Fig. 5B). The paw withdrawal latency (PWL) and the tail flick latency decreased in an intensity-dependent manner (Fig. 5C, Supplementary Fig. 5C and supplementary table 2).

To differentiate nocifensive or non-nocifensive paw withdrawal reflexes, we used a "pain score" system (paw guarding, paw shaking, jumping, and eye grimace) that we previously established with high-speed imaging[16]. Here, we also included *TrpV1^Cre;Advil^FlpO;Rosa^ReaChR* (*TrpV1-ReChR*) mouse as a positive control for nocifensive behaviors. In *Split^Cre-Aβ ReChR* mice, laser at all intensities triggered paw withdrawal reflex (Fig. 5B), but paw withdraw reflexes evoked by 5 mW blue laser did not show any nocifensive features (supplementary movie 4), while 10 mW and 20 mW blue lasers triggered responses with minor nocifensive features (pain score: 0 ± 0 (5 mW), 0.2 ± 0.08 (10 mW), and 0.74 ± 0.13 (20 mW)). In contrast, 5 mW blue laser evoked strong nocifensive behaviors (indicated by high pain scores, 3.52 ± 0.13) in *TrpV1-ReChR* mice (Fig. 5D and supplementary movie 5) as well as the 100% response rate and short PWL (0.21 ± 0.03) (Fig. 5B, C). Interestingly, after CFA-induced inflammation, 5 mW blue laser triggered a significantly higher percentage of paw withdrawal response and a significant shorter PWLs (Fig. 5E, F), indicating that the same stimuli evoked stronger responses of Split^Cre-Aβ-LTMRs in chronic inflammatory nociceptive condition. In addition, 5 mW blue laser triggered responses with significantly higher pain scores at 2 h (*p* = 0.037), days 1 (*p* = 0.0022), 3 (*p* = 0.05), 7 (*p* < 0.0001), 14 (*p* = 0.0005), 21 (*p* = 0.018), and 28 (*p* = 0.017) post CFA injection (Fig. 5G, supplementary movie 6, and supplementary table 2), suggesting that activation of Aβ-LTMRs triggers nociception in chronic inflammatory nociceptive condition.

In addition, we performed conditioned place preference (CPP) assay (Fig. 5H), using littermate control mice (no expression of ReChR) treated with CFA, *Split^Cre-Aβ ReChR* mice treated with saline, *Split^Cre-Aβ ReChR* mice treated with CFA, or *TrpV1-ReChR* mice with no treatment, and paired one side of chamber with plantar optogenetic stimulation (5 mW, 10 Hz). As the positive control, *TrpV1-ReChR* mice (*n* = 4) displayed obvious aversion post-stimulation (−68.97 ± 4.39 %). Interestingly, *Split^Cre-Aβ ReChR* mice with CFA but not the two control groups showed a significant increase of aversion toward the blue laser-paired chamber (*Split^Cre-Aβ ReChR*, Saline = −0.77 ± 8.69 % vs. *Split^Cre-Aβ ReChR*, CFA = −41.88 ± 8.3 %, *p* = 0.004;

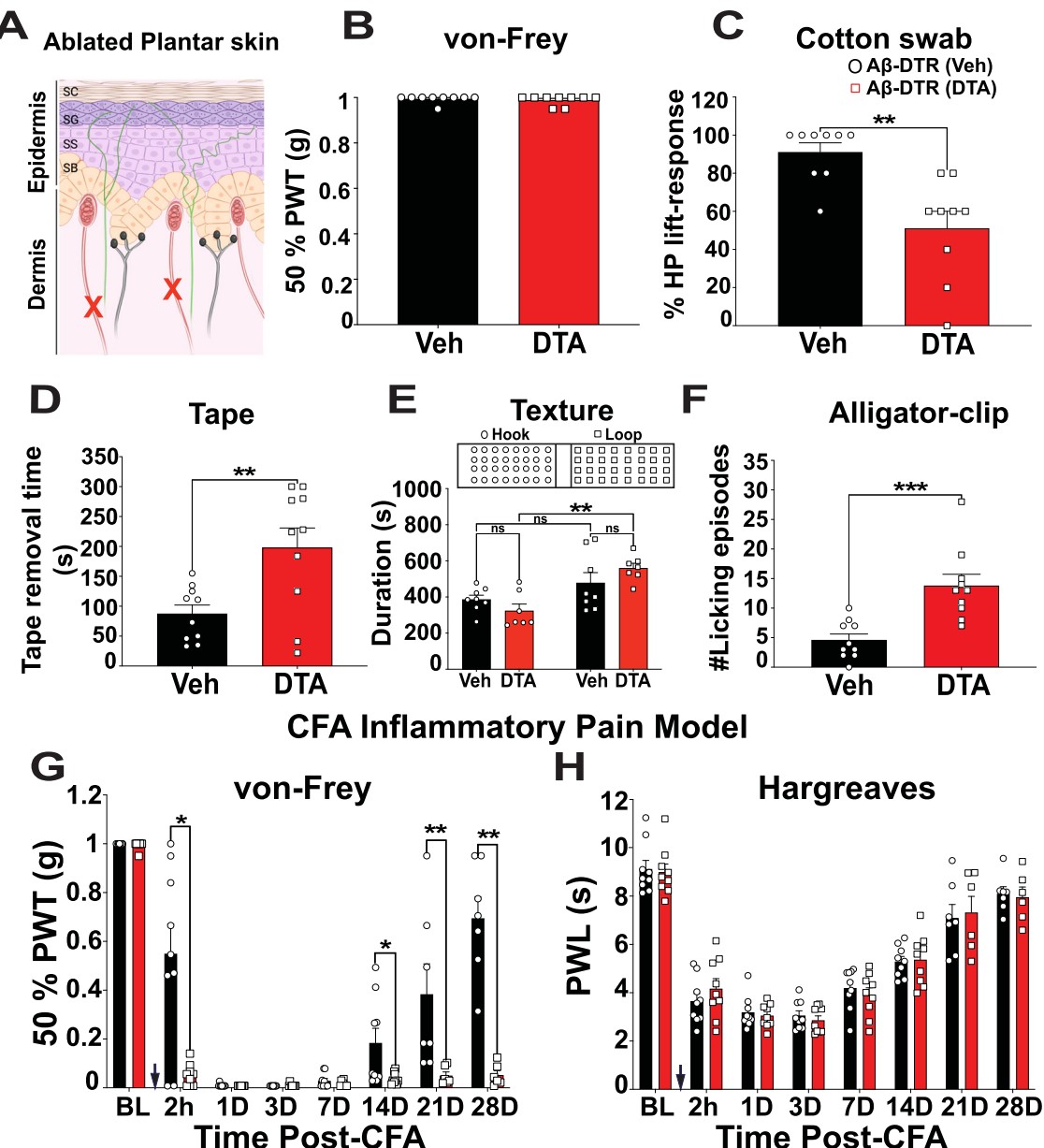

**Fig. 4 | Ablation of Split^Cre-Aβ-LTMRs reduced gentle touch sensation but increased mechanical nociception in the glabrous skin. A** Illustration showing cutaneous sensory afferents of the ablated plantar skin; green fibers represent nociceptive C and Aδ fibers; red fibers represent RA Aβ-LTMRs innervating Meissner's corpuscles (red capsules); black fibers represent SA Aβ-LTMRs innervating Merkel cells; 'X' symbol represents the DTA-induced ablation. **B** Ablation of Split^Cre-Aβ-LTMRs did not affect 50% PWT of the von Frey hair test ($p > 0.99$), $n = 9$ mice in each group. **C** Percentage of paw lifting responses to dynamic cotton swab was significantly attenuated in the ablated mice ($p = 0.0014$), 5 trials/mouse, $n = 9$ mice in each group. **D** The tape removal time significantly increased in the ablated mice ($p = 0.0063$), $n = 10$ mice in each group. **E** Chamber preference of different floor texture was altered in the ablated mice ($p = 0.001$) compared to the control mice ($p = 0.097$). $n = 8$ for control and 7 for the ablated mice. **F** The number of licking

episodes with the application of an alligator clip significantly increased in the ablated mice ($p = 0.0002$), $n = 10$ mice in each group. **G** The ablated mice showed significantly decreased 50% PWT in chronic inflammation condition at 2 hour ($p = 0.0149$), 14th day ($p = 0.0427$), 21st day ($p = 0.0012$), and 28th day ($p = 0.0012$ g) in comparison to vehicle-treated mice. $n = 9$ mice per group except the days 21 and 28 (vehicle: $n = 7$ mice and DTA: $n = 6$ mice). Black arrow indicates the time of intraplantar injection of CFA. **H** The ablated mice showed no significant change in the paw withdrawal latency (PWL) of the Hargreaves test in chronic inflammation condition (vehicle and DTA: $n = 9$ mice). Black and red color of the bars represent vehicle and DTA treatment respectively. Error bars represent Mean ± S.E.M. Unpaired, two-tailed Mann-Whitney test. *$p < 0.05$, **$p < 0.01$; ***$p < 0.001$, ns = non-significant. Source data are provided as a Source Data file.

Littermate control, CFA = −6.44 ± 8.54 % vs. *Split^Cre-Aβ ReaChR*, CFA = −41.88 ± 8.3 %, $p = 0.029$) (Fig. 5I). These behavioral results indicate that local optogenetic stimulation of Split^Cre + Aβ-LTMRs in inflammatory nociceptive condition induced aversion.

Activation of Aβ-LTMRs was also tested in chronic neuropathic pain using the medial plantar nerve ligation (MPNL) model[32]. Similarly, 5 mW blue laser triggered a significantly higher percentage of paw

withdrawal response, significantly reduced PWLs (Fig. 5J,K and supplementary table 2), and significantly increased pain scores at days 1 ($p = 0.0292$), 3 ($p < 0.0001$), 7 ($p < 0.0001$), 14 ($p < 0.0001$), 21 ($p < 0.0001$) and 28 ($p < 0.0001$) post-surgery (Fig. 5L and supplementary table 2). Together, our results suggest that local activation of Aβ-LTMRs triggers nociception in both chronic inflammatory and neuropathic pain conditions.

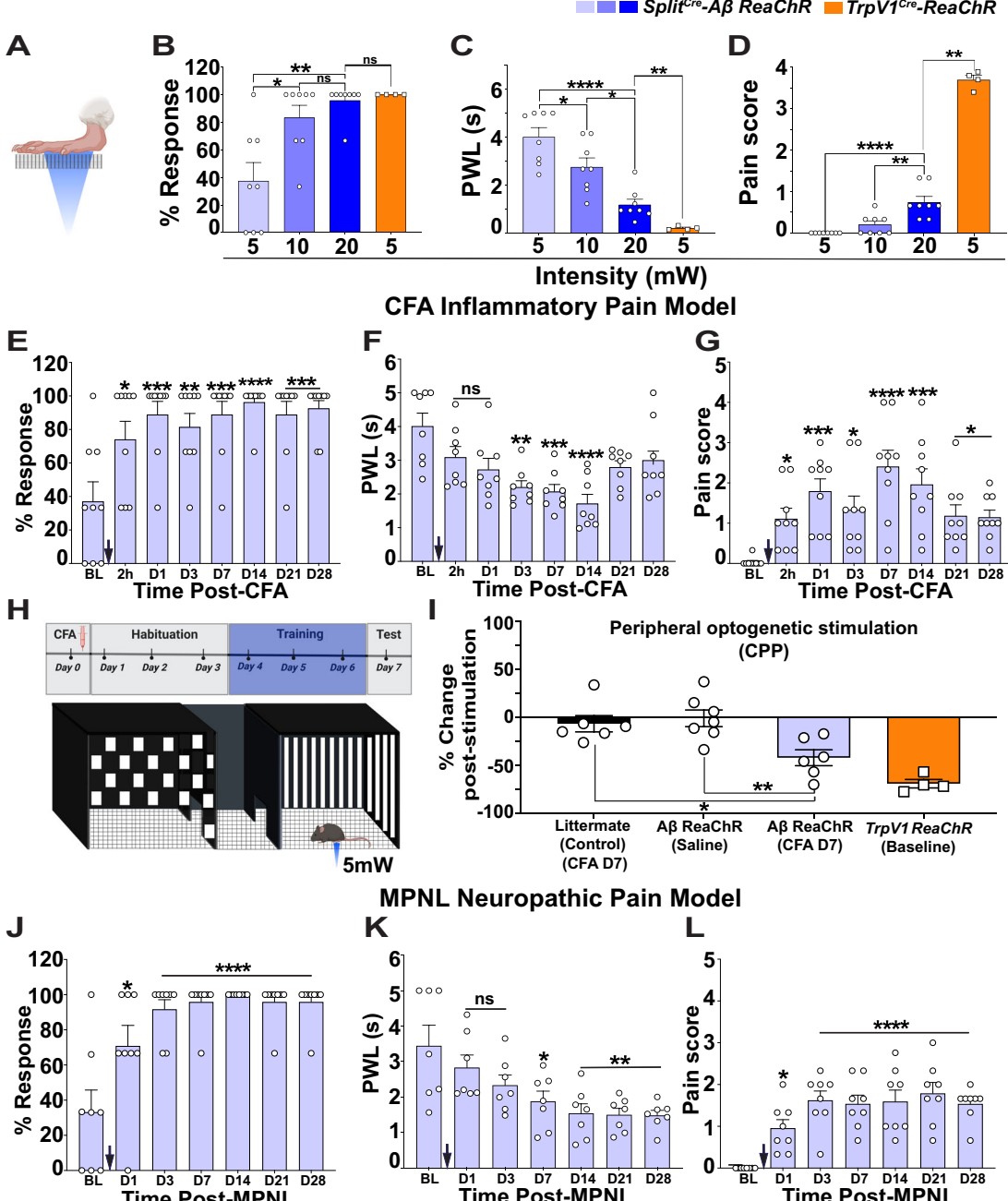

**Fig. 5 | Local optogenetic activation of Split^Cre-Aβ-LTMRs evoked nociception in mouse chronic pain models. A** Illustration showing peripheral optogenetic activations. The percentage of paw withdrawal (PW, (**B**)), PWL (**C**), and pain score (**D**) of *Split^Cre-ReaChR* mice (*n* = 8) at 5, 10, and 20 mW blue laser stimuli and *TrpV1^Cre-ReaChR* mice (*n* = 4). **E** 5 mW blue laser triggered significantly higher percentages of PW in CFA model at all time-points post-CFA. **F** 5 mW blue laser significantly decreased PWLs at days 3, 7, and 14. *n* = 8 mice in each group. **G** 5 mW blue laser triggered significantly higher pain scores at all time-points post-CFA. *n* = 9 mice in each group (**E**, **G**). **H** Illustration of the CPP paradigm. **I** CFA-treated *Split^Cre-ReaChR* mice (*n* = 6) showed significantly higher percentage of aversion in comparison to the CFA-treated littermate control (*n* = 6) and saline-treated *Split^Cre-ReaChR* mice (*n* = 7). *TrpV1-ReaChR* mice (*n* = 4) showed a high aversion. **J** 5 mW blue laser

triggered significantly higher percentages of PW in a MPNL model at all time-points post-MPNL. **K** 5 mW blue laser significantly decreased PWLs at days 7, 14, 21, and 28. *n* = 7 mice in each group. **L** 5 mW blue laser triggered significantly higher pain scores at all time-points post-MPNL. *n* = 8 mice in each group for (**J**, **L**). Light blue, medium blue and dark blue bars represent 5, 10 and 20 mW blue light stimuli to *Split^Cre-ReaChR* mice respectively. Orange bar represents 5 mW blue light stimuli to *TrpV1^Cre-ReaChR* mice. Black arrows in Figures (**E**–**G**) and (**J**–**L**) represent the onset of CFA-induced inflammatory and MPNL neuropathic pain models respectively. Error bars represent Mean ± S.E.M. See source data file for detailed p values and statistical tests. **p* < 0.05, ***p* < 0.01, ****p* < 0.001, *****p* < 0.0001. Source data are provided as a Source Data file.

## Changes in electrophysiological properties of Split^Cre-ReaChR+ Aβ- LTMRs following CFA-induced inflammation

To determine changes in electrophysiological properties of Split^Cre-ReaChR+ Aβ-LTMRs in chronic nociceptive condition, we performed opto-tagged recording of these afferents in saline- or CFA-treated mice

at post-injection day 7. Mechanically-evoked RA impulses were recorded from light-sensitive (ReaChR/YFP tagged) afferent fibers that innervate the glabrous skin of hind paws (Fig. 6A). We detected RA impulses of light-sensitive afferent fibers in both saline- (*n* = 15 units) and CFA-injected (*n* = 5 units) mice. The number of impulses increased

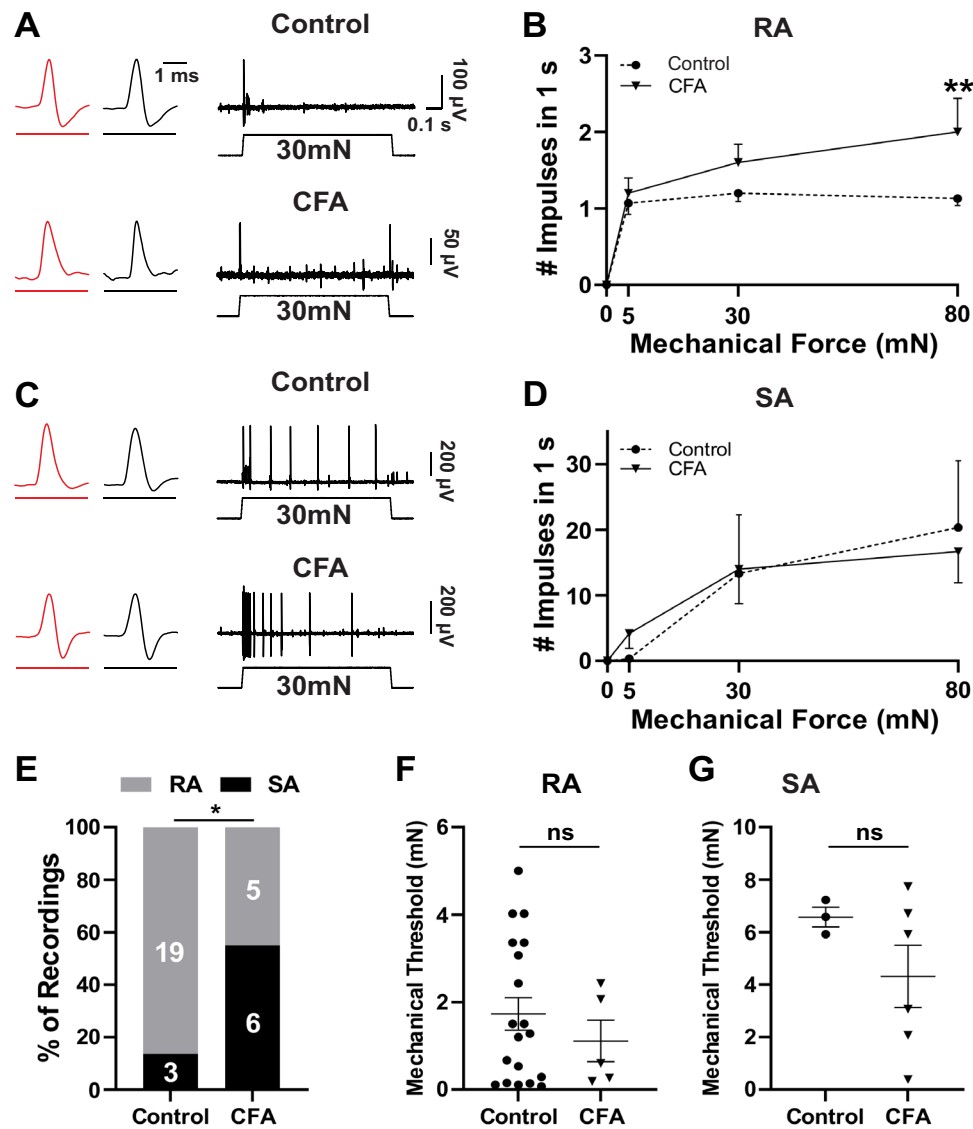

**Fig. 6 | Split^Cre-ReaChR^+ Aβ-LTMRs in the hind paw glabrous skin fire more action potentials in response to mechanical forces after CFA treatment. A** Two sample traces showing mechanically evoked RA impulses recorded from light sensitive (ReaChR/YFP tagged) afferent fibers that innervate the glabrous skin of hind paws. Top, Saline-injected paw (control), Bottom, CFA-injected paw. RA impulses were evoked by tissue indentation at 30 mN at the receptive field where red LED light evoked impulses. Sample traces at expanded time scale show light-evoked (red) and mechanically evoked AP impulses (black). **B** Summary data of RA impulses evoked by tissue indentation at 0, 15, 30, and 80 mN in light-sensitive afferent fibers of saline-injected group ($n = 15$) and CFA-injected group ($n = 5$, $p = 0.0016$ at 80 mN). **C** Similar to (**A**) except SA impulses were recorded from a saline-injected paw (top, control) and CFA-injected paw (bottom). **D** Similar to (**B**)

except SA impulses were recorded from the control group ($n = 3$) and CFA-injected group ($n = 6$). **E** Percent of light-sensitive afferent fibers that show RA ($n = 19$) or SA impulses in control group ($n = 3$), and RA ($n = 5$) or SA impulses ($n = 6$) in CFA group ($p = 0.013$), *$p < 0.05$, two-sided Chi's square test. Grey and black bars show RA and SA recordings respectively. **F** Thresholds of RA-LTMRs in control ($n = 19$) and CFA ($n = 5$) groups. **G** Thresholds of SA-LTMRs in control ($n = 3$) and CFA ($n = 6$) groups. In (**D**, **G**), control group was pooled from both saline-injected and un-injected paws, 2 saline-injected and one un-injected paws for SA in (**D**, **E**, **G**); 15 saline-injected and 4 un-injected paws for RA in (**E**, **F**). In (**B**, **D**, **F**, **G**), data represent Mean ± S.E.M., **$p < 0.01$, two-way ANOVA with post hoc Bonferroni's test, or ns, not significantly different, two-sided Student's *t*-test. Source data are provided as a Source Data file.

proportionally with higher mechanical force in CFA-treated groups, which was significant at 80 ($p < 0.01$) mN force (Fig. 6B). Similarly, we detected tissue indentation induced SA impulses in light-sensitive afferent fibers of saline- ($n = 3$ units) and CFA-injected ($n = 6$ units) mice (Fig. 6C), which didn't display obvious changes of impulse numbers in responses to different mechanical forces (Fig. 6D).

The number and percentages of opto-tagged afferents showing RA and SA responses were quantified (Fig. 6E). In the glabrous skin of the control mice, few light-sensitive afferents were SAs ($n = 3$), and most of them were RAs ($n = 19$), similar to the result of naïve animals (Fig. 2H). However, with CFA-induced inflammation, the percentage of RA afferents ($n = 5$) decreased while that of SA ($n = 6$) increased

significantly ($p < 0.05$) (Fig. 6E). Mechanical thresholds of RA and SA afferents in control and CFA groups were similar (Fig. 6F, G). Overall, these data suggest that in chronic inflammation, the electrophysiological properties of some RA Aβ-LTMRs became SA-like, which could be one of the underlying mechanisms for triggering stronger local activity and inducing mechanical hyperalgesia in the CFA model.

To determine whether ReaChR^+ LTMRs increased firing was due to an alteration in peripheral end organ innervation, we performed foot pad section immunostaining for ReaChR^+ afferents, Merkel cells (Supplementary Fig. 5D, E), or Meissner's corpuscles (Supplementary Fig. 5F, G) using control or CFA-treated Split^Cre-Aβ ReaChR mice at post-CFA Day 7. We found that the number of Merkel cells per foot pad skin

section showed no significant difference between groups (control = 13.02 ± 2.58 vs. CFA = 9.45 ± 1.53, $p = 0.18$) (Supplementary Fig. 5H), nor did the percentage of GFP$^+$ fibers innervating Merkel cells per foot pad section (control = 10.38 ± 2.37% vs. CFA = 17.21 ± 2.95%, $p = 0.088$) (Supplementary Fig. 5I). The number of Meissner's corpuscles per foot pad section showed a non-significant decrease trend in CFA-treated mice (control = 4.496 ± 0.51 vs. CFA = 2.89 ± 0.60, $p = 0.093$) (Supplementary Fig. 5J), but the percentage of GFP$^+$ fibers innervating Meissner's corpuscles per foot pad section significantly decreased in CFA-treated mice (control = 69.42 ± 7.87% vs. CFA = 46.17 ± 5.37%, $p = 0.041$) (Supplementary Fig. 5K). Thus, our results suggest that some Split$^{Cre}$-ReaChR$^+$ afferents retract from Meissner's corpuscles in the inflammatory paw. This structural alternation, loss of corpuscles[33], and transcriptomic changes induced by inflammation[34] might contribute to the increased firings of Split$^{Cre}$-ReaChR$^+$ RA afferents in the CFA model.

### Dorsal column stimuli of Split$^{Cre}$-ReaChR$^+$ Aβ-LTMRs activated spinal cord dorsal horn inhibitory neurons

How could both global ablation and local activation of Aβ-LTMRs promote mechanical hyperalgesia? We speculated that the mechanism came from the unique morphologies and circuits associated with Aβ-LTMRs. In contrast to nociceptors, which have central terminals innervating only one to two spinal cord segment[35], Aβ-LTMRs have ascending and descending axons projecting through the dorsal column and 3$^{rd}$ order collaterals covering the dorsal horn of 6 to 8 spinal cord segments (3 to 4 segments along both ascending and descending axons)[36,37]. This anatomical feature allows an Aβ-LTMR to "gate" nociceptors (inter-modality crosstalk) or other Aβ-LTMRs residing in 3 to 4 dermatomes away (inter-somatotopy crosstalk) (see below). Thus, even though locally affected Aβ-LTMRs are sensitized to trigger nociception in inflammation or nerve injury regions, the overall effect of all Aβ-LTMRs, most of which are unaffected, is still to inhibit mechanical nociception.

If our model is correct, then the global activation of Aβ-LTMRs, even in chronic nociceptive model, theoretically should still inhibit mechanical hyperalgesia. To test this idea, we implanted light-cannula above the T11 dorsal column of the spinal cord of Split$^{Cre}$-Aβ ReaChR mice (Fig. 7A). Spinal light stimulation (15 minutes total, 0.5 mW or 10 mW, 30 second on (10 Hz) and 1 minute off, 10 cycles) triggered some spontaneous behaviors (supplementary table 3), but they calmed back to the resting state when light was off. We then did VFH test to measure 50% PWT at different time points after the light off (Fig. 7A). We averaged PWT of both paws for quantification as the light cannula was implanted at the dorsal column and would affect both sides of the spinal cord. The 0.5 mW but not 10 mW blue laser spinal cord stimuli did not significantly alter the PWT afterwards (Fig. 7B and supplementary table 3). Thus, it was chosen to test functions of dorsal column activation of Aβ-LTMRs in chronic inflammatory nociception.

The T11 dorsal column stimuli should activate a large population of Aβ-LTMRs, whose ascending and descending axons primarily projected through the dorsal column, and their downstream neurons in the spinal cord dorsal horn (DH). If so, DH interneurons at the lumbar enlargement, where hind paw mechanical hyperalgesia are transmitted, would also be affected. To test this idea, we performed c-Fos immunohistochemistry using T11 and L4 spinal cord sections 90 minutes after dorsal column stimuli using 0.5 mW blue laser (Supplementary Fig. 6A–L). We found that the number of c-Fos$^+$ cells significantly increased at both T11 and L4 spinal cord DH with light stimulation of Split$^{Cre}$-Aβ ReaChR mice, in comparison to no light stimuli control mice. Specifically, a significant increase in the number of c-Fos$^+$ neurons was found in deeper layers (T11: 4.1 ± 0.4 (No light), 16.1 ± 1.0 (Light), $p < 0.0001$; L4: 3.5 ± 0.3 (No light), 16.3 ± 1.3 (Light), $p < 0.0001$) but not in superficial layers (T11: 6.2 ± 0.6 (No light), 9.1 ± 1.3 (Light), ns; L4: 7.4 ± 0.4 (No light), 9.8 ± 1.5 (Light), ns) (Fig. 7C).

To further discern whether the activated DH neurons are excitatory or inhibitory, we performed RNAscope in situ hybridization of cFos, Slc17a6 (Vglut2) and Slc32a1 (Vgat) (Fig. 7D and Supplementary Fig. 6M–P). Around 80% cFos+ neurons were Slc32a1$^+$ (Fig. 7E). Together, our results suggested that dorsal column stimuli of Split$^{Cre}$-ReaChR$^+$ Aβ-LTMRs would preferentially activate DH inhibitory interneurons in a broad range of spinal cord levels.

### Dorsal column activation of Split$^{Cre}$-ReaChR$^+$ Aβ-LTMRs alleviated mechanical hyperalgesia of chronic inflammatory pain

Lastly, we tested whether dorsal column stimuli of Split$^{Cre}$-ReaChR$^+$ Aβ-LTMRs using 0.5 mW blue laser attenuated or enhanced nociception in CFA-induced chronic inflammatory pain model. One hindpaw of dorsal column implanted Split$^{Cre}$-ReaChR mice were injected with CFA. The same light stimuli (0.5 mW) and VFH tests were performed at different time points after CFA treatment (Fig. 7A). Overall, we found a significant alleviation of mechanical hyperalgesia five to thirty minutes after dorsal column activation of Aβ-LTMRs. This lasting effect could be related to optogenetic stimuli induced synaptic potentiation[38,39]. Specifically, a significantly reversal of the PWT was observed at 2 hours, 7 days, 14 days post-CFA (2 hours: 0.13 ± 0.026 g (pre-stimulation), 0.618 ± 0.069 g (5 min), $p < 0.0001$; 0.633 ± 0.077 g (30 min), $p < 0.0001$) (Fig. 7Fi); (day 7: 0.008 ± 0 g (prestim), 0.458 ± 0.062 g, (5 min), $p = 0.0082$; 0.649 ± 0.091 g (30 min), $p = 0.0093$) (Figure 7Fii); (day 14: 0.026 ± 0.006 g (prestim), 0.663 ± 0.094 g (5 min), $p = 0.011$; 0.676 ± 0.089 g (30 min), $p = 0.0079$) (Figure 7Fiii). At days 21 post-CFA, as mechanical hyperalgesia started to recover, no significant change of PWT was observed (Figure 7Fiv). These results provide direct evidence to support that global activation of Aβ-LTMRs could attenuate mechanical hyperalgesia in chronic pain.

## Discussion

In this study, we combined intersectional mouse genetics, opto-tagged electrophysiology recordings, and high-speed imaging behavioral assays to clarify roles of Aβ-LTMRs in transmitting and alleviating mechanical hyperalgesia. Our results revealed that both global ablation and local activation of Aβ-LTMRs promoted mechanical hyperalgesia, whereas their global activation alleviated it. Therefore, we propose a model (Fig. 8), which integrates the inter-modality crosstalk between the nociceptive pathway and Aβ-LTMRs and the inter-somatotopy crosstalk among Aβ-LTMRs at different spinal segments, to explain the complicated phenotypes of Aβ-LTMRs in mechanical hyperalgesia. Our model suggests that the global activation plus local inhibition of Aβ-LTMRs would be an effective strategy for treating mechanical hyperalgesia.

A half century after the introduction of Gate Control Theory[5], the exact functions of Aβ mechanoreceptors in chronic pain are still not fully resolved. The primary function of these afferents is mediating discriminative touch and tactile/vibration sensation. However, these neurons also play important roles in "gating" or inhibiting other somatosensory pathways to generate the appropriate sensation. The fast conduction velocity of Aβ-LTMRs put them in a great position for this role. In response to a mechanical stimulus, signals of Aβ-LTMRs arrive at the dorsal spinal cord first and "set up" the "gate" for other mechanosensory pathways with slower conduction velocities (Aδ- and C-LTMRs and high-threshold mechanoreceptors). Thus, activating Aβ-LTMRs has been a commonly sorted strategy for treating chronic pain[13,40]. On the other hand, multiple studies also suggest that Aβ-LTMRs mediate mechanical allodynia in chronic pain conditions[7,8,28,41,42]. How to consolidate these experimental results?

Here we utilized intersectional genetics to selectively ablate Aβ-LTMRs or optogenetically activate them either locally at the affected site (peripheral) or globally at the dorsal column (central) and examined behavioral outcomes in baseline and chronic pain conditions. Interestingly, activities of locally affected Aβ-LTMRs in chronic

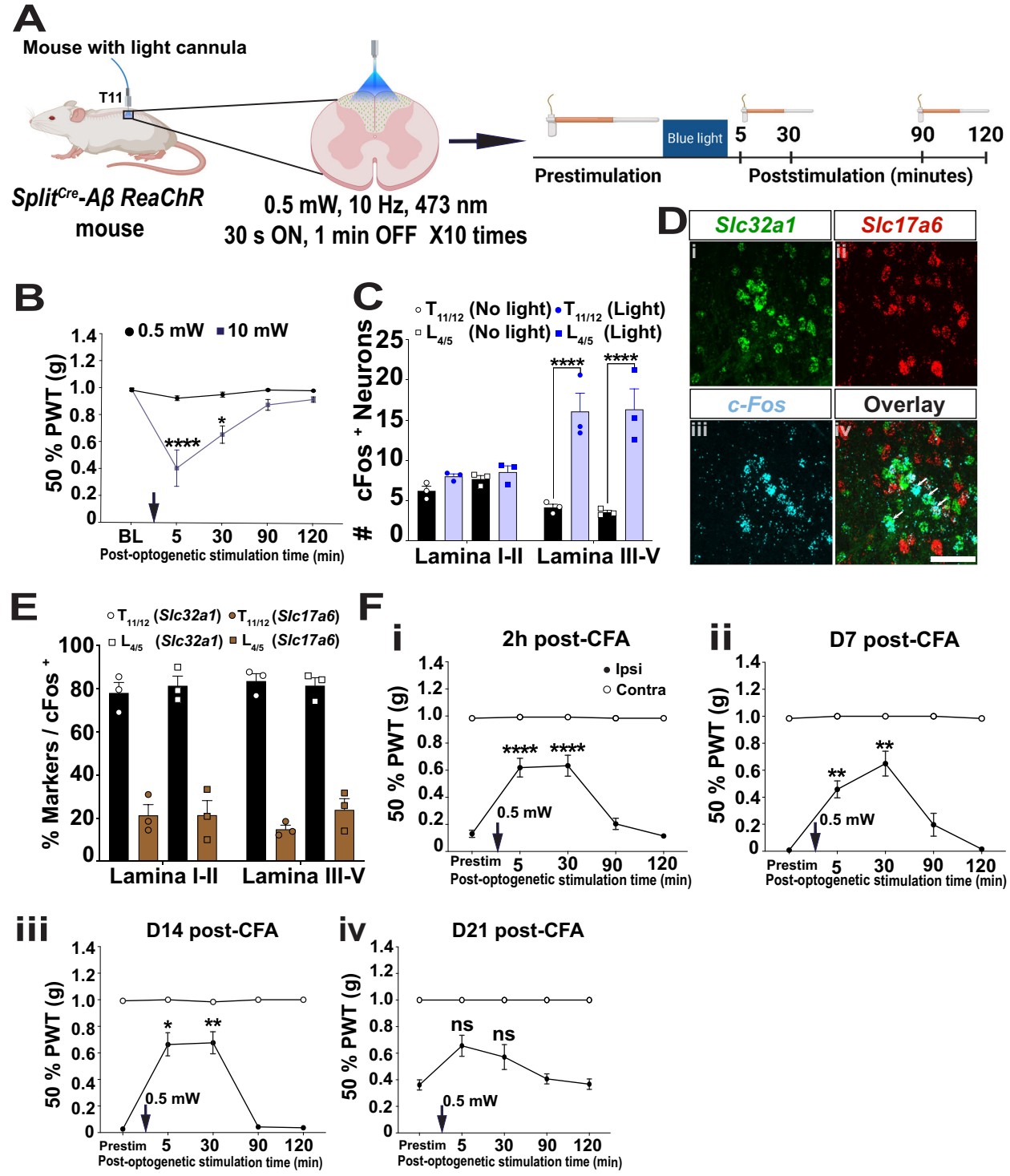

**Fig. 7 | Dorsal column activation of Split^Cre-ReaChR+ Aβ-LTMRs alleviated mechanical hyperalgesia in the chronic inflammatory pain model. A** Illustration showing our experimental paradigm. Blue laser stimulates ascending dorsal horn axons of Split^Cre- ReaChR+ Aβ-LTMRs (green color). VFH tests were performed at the baseline and different time points of the CFA model. **B** PWT of spinal cord stimulation using 0.5 and 10 mW blue laser. $n = 7$ mice. Black arrow indicates the time of blue light stimulation. **C** Quantification of c-Fos+ neurons in the superficial and deep dorsal horn laminae of the thoracic and lumbar spinal cord of no-light and light- stimulated Split^Cre-Aβ ReaChR mice (without treatment). Light blue and black bars represent light (0.5 mW) and no light conditions respectively. **D** RNAscope of c-Fos, Slc17a6, and Slc32a1 with lower thoracic spinal cord section. Scale bar = 50 μm. **E** Quantification of excitatory (Slc17a6+, brown bar) and inhibitory (Slc32a1+, black bar) interneurons co- labelled with c-Fos in the superficial and deep dorsal horn laminae of the thoracic and lumbar spinal cord after light stimulation. $n = 3$

mice, 4 sections per mouse. **F** Global activation of Split^Cre-ReaChR+ Aβ-LTMRs alleviated mechanical hyperalgesia in mice with CFA-induced chronic inflammatory pain. i. Spinal stimulation at 2 hour post-CFA significantly reversed the PWT at 5 min and 30 min post-light stimulation. ii. At D7 post-CFA, a significant reversal of PWT was observed at 5 min. and 30 min.post-light stimulation. iii. At D14 post-CFA, a significant reversal of PWT was observed at 5 min and 30 min. post-light stimulation. iv. At D21 post-CFA, no significant difference was observed in PWT post-light stimulation. Black arrow indicates the time of blue light stimulation. $n = 6$ mice except panel C (3 mice, 5-7 sections per mouse for statistical comparison) and E (3 mice, 3–4 sections per mouse). Error bars represent Mean ± S.E.M. Two-way ANOVA followed by Bonferroni test. *$p < 0.05$, **$p < 0.01$, ****$p < 0.0001$, ns = non-significant. See source data file for detailed $p$ values and statistical tests. Source data are provided as a Source Data file.

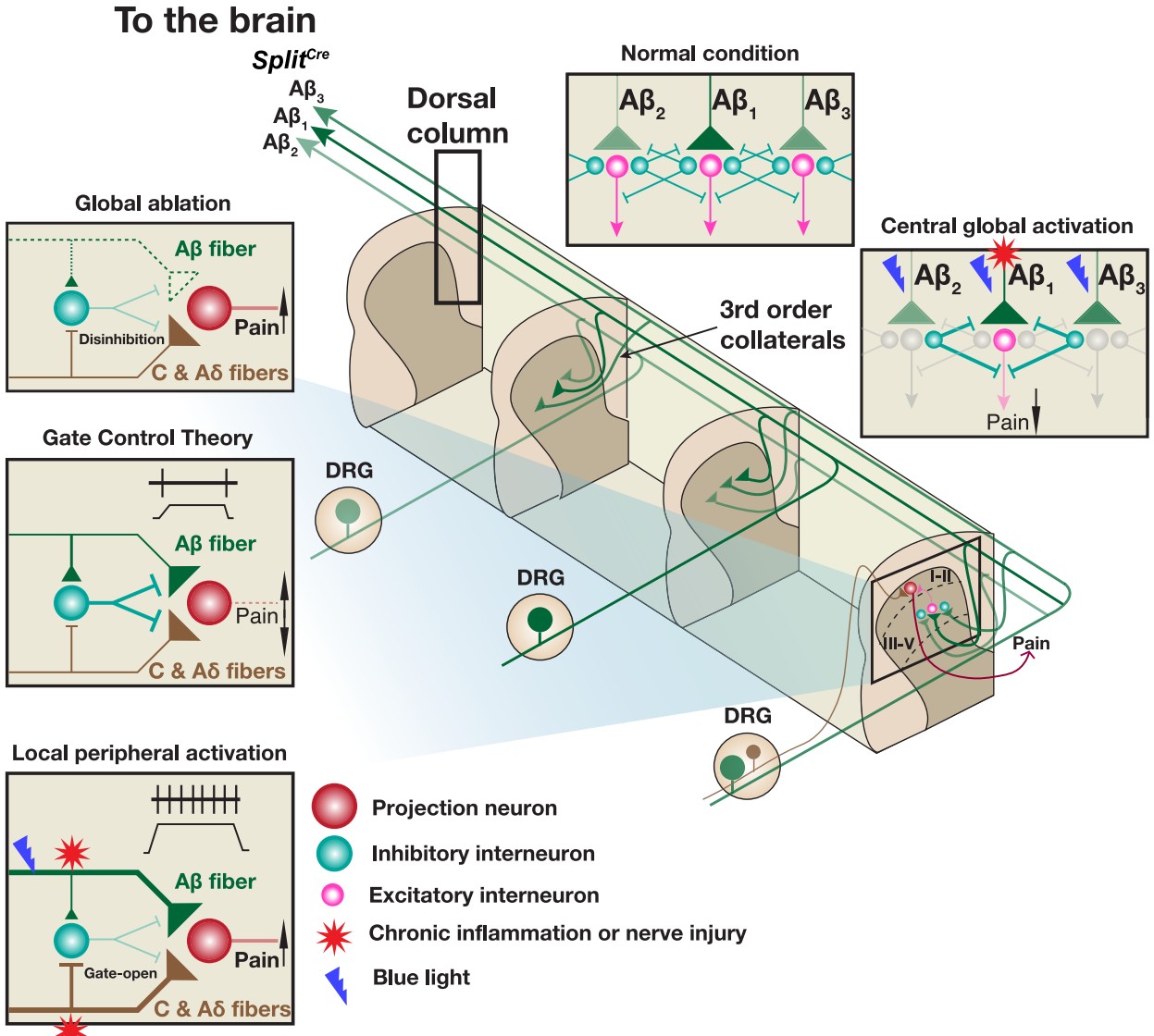

**Fig. 8 | A model to explain how Aβ-LTMRs function in both transmitting and alleviating mechanic hyperalgesia.** Illustration shows anatomy of central projections of nociceptors and Aβ-LTMRs and the inter-modal and inter-somatotopic crosstalk. Central terminals of nociceptors only innervate one or two spinal cord segments, whereas Aβ-LTMRs have ascending and descending axons projecting through the dorsal column and 3rd order collaterals covering the dorsal horn of 6 to 8 spinal cord segments. This anatomical feature allows an Aβ-LTMR to interact with nociceptors and other Aβ-LTMRs in 3 to 4 dermatomes away.

inflammation triggered nociception (Fig. 5). We found that RA mechanoreceptors in the glabrous skin following tissue inflammation fired more numbers of action potentials (displaying a SA-like property) (Fig. 6). This idea may need further experimental tests in future studies. On the other hand, global activities of Aβ-LTMRs, many of which were unaffected, still inhibited transmission of mechanical nociception in both acute and chronic pain conditions in a modality-specific manner (Figs. 4, 7). This distinctive "local" vs. "global" functions of Aβ-LTMRs in mechanical nociception, as explained by our model (Fig. 8), will help to consolidate the existing data in this topic and provide a theoretic reference for the design of prospective treatment strategies.

Our findings are somewhat different from two previous studies[9,43] in this topic, which used different genetic tools for ablation or optogenetic activation. The first study found that DTA ablation of Trkb+ afferents attenuated mechanical allodynia in a neuropathic pain (SNI) but not inflammatory pain (CFA) models (Dhandapani et al., 2018). A main difference between ours and this study is the genetic tools, which manipulated largely non-overlapping sensory afferents. We used *Split-Cre* allele that preferentially recombined in Ret+ Aβ-RA-LTMRs (~80%)

and some Aβ-SA-LTMRs in both glabrous and hairy skin (Fig. 1 and Supplementary Figs. 1, 2), whereas the other study used *TrkB-CreERT2* mice, which preferentially label Aδ-LTMRs (D-hair) in the hairy skin[17] and a few Aβ-RA-LTMRs in the glabrous skin[44]. For this study, a high percentage (~75%) of paw Meissner corpuscles were innervated by *Split-Cre* labeled afferents, and ~62% of these innervations was lost in the ablated mice (Fig. 3J), and DTA-ablated mice displayed obvious deficits in dynamic gentle mechanical force and tactile sensation (Fig. 4). In contrast, the study by Dhandapani et al. found no mechanosensory deficits from paws of ablated mice at the baseline condition. Although both Ret+ and Ntrk2+ Aβ-RA-LTMRs innervate Meissner corpuscles, they make two distinct subtypes[44]. Thus, the global DTA ablation using *TrkB-CreERT2*, in which Aδ-LTMRs and Ntrk2+ Aβ-RA-LTMRs were disrupted, and *Split-Cre* lines, in which Ret+ Aβ-RA-LTMRs and some Aβ-SA-LTMRs were disrupted, generated different behavioral outcomes.

The second study used a *VGlut1-Cre* mouse line, which should recombine in Aβ-LTMRs as well as other large-diameter DRG neurons. This study didn't detect behavior changes when optogenetically

activating VGlut1$^{Cre}$-Aβ-LTMRs in a neuropathic pain model[9]. As the authors discussed in that paper, it is hard to differentiate between touch-like and pain-like pain reflex behaviors without a high-speed camera. This is what we used in this study to improve resolution of behavior assays.

Multiple existing techniques, presumed to involve the manipulation of Aβ-LTMRs, are used in practice for treating chronic pain. The best known one is the SCS, which was developed based on the "gate control theory" and was designed to target Aβ-LTMR axons projecting through the dorsal column[12]. The SCS is effective for treating different chronic pain conditions, refractory chronic pain, and even those failed available pharmacological approaches[45,46]. Since several types of axons, besides those of Aβ-LTMRs, project through the dorsal column, and can simultaneously be activated by SCS, whether stimulating Aβ-LTMR axons alone in the dorsal column is sufficient for alleviating chronic pain has remained untested. Our study provides direct supportive evidence for the chronic pain-alleviating effect of activating Aβ-LTMRs at the dorsal column (Fig. 8). Nevertheless, our results do not exclude contribution of other dorsal column fibers in this effect. Some Transcutaneous Electrical Nerve Stimulation (TENS) strategies aim to target peripheral Aβ-LTMRs 2-3 dermatomes away from the injury site, which effects in alleviating pain were supported by several clinic trials[47–50]. These practices and effects also align well with our results and model. Finally, our model proposes a potentially more effective, compound, strategy to target Aβ-LTMRs for treating mechanical hyperalgesia and chronic pain: activating them globally or 3-4 dermatomes away from the injury site and combining with local inhibition of the affected Aβ-LTMRs.

## Methods
The details of reagents and software used in this study can be found in Supplementary Source Data 2.

### Animals
*Advil*$^{FlpO}$[20] and *Split*$^{Cre}$[17,18] mice were imported from Dr. David Ginty's lab at the Harvard University, and the *Tau*$^{dsDTR}$[26] mouse line was imported from Dr. Martyn Goulding's lab at the Salk Institute. Other mice (*Rosa*$^{ReaChR}$, strain #:024846; Ai9, strain #:007905, *TrpV1*$^{Cre}$, strain #:017769, and C57BL/6J, strain #:000664) were purchased from Jackson's laboratory. Animals (6–20 weeks old) were housed in facilities at the University of Pennsylvania and at the University of Alabama at Birmingham at controlled room temperature (20–23 °C) and a humidity level between 30–70 % in a standard 12-h light/dark cycle, with water and food pellets available *ad libitum*.

All experiments were conducted in accordance with the National Institute of Health guidelines and with approval from the Institutional Animal Care and Use Committee of University of Pennsylvania and University of Alabama at Birmingham. Both male and female mice were used in all experiments.

### Immunohistochemistry of cryosections
Mice were anesthetized using Ketamine/xylazine/acepromazine cocktail and transcardially perfused with 4% PFA in phosphate buffer solution (PBS). Lumbar spinal columns, DRGs, and skin pieces were dissected and post-fixed for 2–4 hours in 4% PFA in PBS at 4 °C, cryoprotected in 30% sucrose in PBS O/N at 4 °C, and embedded in OCT. 20 μm cryosections of spinal cord, DRG and skin were cut using a cryostat (Leica CM1950). DRG sections were collected on Superfrost™ Plus slides (Fisherbrand) and allowed to dry at room temperature overnight, while spinal cord and skin sections were collected in multi-well plate and 1.5 mL Eppendorf tubes (Skin sections) for floating section immunostaining. Slides or floating sections were washed in PBS containing 0.2% TritonX-100 (3 × 10 minutes) and then blocked in PBS containing 5% lamb serum and 0.2% TritonX-100 (PBT) for 1 hour at room temperature. Primary antibodies were diluted in the same buffer,

incubated O/N at 4 °C, and then washed in PBT (3 × 10 minutes). Secondary antibodies were incubated in blocking buffer at 1:500 dilution for one hour at room temperature. Slides or floating sections were then washed in PBT (3 × 10 minutes), mounted with Fluormount and cover-slip, and sealed using clear nail-polish. Primary antibodies used include chicken anti-GFP (1:1000; Aves, GFP-1020), rabbit anti-CGRP (1:1000; Immunostar, 24112), rabbit anti-NF200 (1:1000; Sigma, N4142), chicken anti-NF200 (1:1000, Aves labs, NFH-3-1003), guinea pig anti-VGlut1(1:1000; Millipore, AB5905), rat anti-K8 (Troma-1) (1:100; Univ of Iowa/DSHB), rabbit anti-Ret (1:100, Immuno-Biological Laboratories, 18121), rabbit anti-parvalbumin (1:200, Swant, PV27), rabbit anti-S100 (1:200, Abcam, ab34686), rabbit anti-cFos (1:800; Cell Signaling Technology, 2250), and Alexa 488 or 594 conjugated IB4 (1:500; Invitrogen, I21411). Secondary antibodies used are Alexa 488, Alexa 594, Cyan or Alexa 647 conjugated goat anti-rabbit antibody, Alexa 647 conjugated goat anti-chicken antibody, and Alexa 647 conjugated goat anti-guinea pig antibody. Secondary antibodies were purchased from either Invitrogen or the Jackson Immunoresearch.

### Mouse spinal cord vibratome sections and c-Fos immunostaining
The spinal cord tissues were vibratome sectioned for c-Fos immunostaining. In brief, after the transcardiac perfusion of mice in 4% PFA, their lower thoracic (T11) and lumbar (L4/5) spinal columns were collected and post fixed in 4% PFA for 1 h at 4 degree. After 3 × 5 min 1X PBS washes, the spinal segments were embedded in 2% low-melt agarose. Using a vibratome machine (VT1200S, Leica Microsystems, Nussloch, Germany), 40 μm thickness spinal cord sections were cut. Sections were washed in PBS containing 0.5% TritonX-100 (3 × 15 minutes), and then blocked in PBS containing 5% lamb serum and 0.5% TritonX-100 (PBT) for 1 hour at room temperature. Rabbit anti-c-Fos (1:800; Cell Signaling Technology) was diluted in the same buffer (1:800), and incubated for ~24 h at 4 °C, then washed in PBT (3 × 15 minutes). Secondary antibodies were incubated in blocking buffer at 1:500 dilution for 90 min at room temperature. Tissues were then washed in PBT (3 × 15 minutes), followed by clearing in 50 and 75% glycerol in PBS (15 min each). Sections were mounted in 75% glycerol, and cover slip was sealed using clear nail-polish (Fisher Scientific, NC1849418).

### Whole mount immunostaining of skin samples
Whole-mount immunohistochemistry of skin was performed as previously described[17]. Mice were anesthetized and transcardially perfused using 4% PFA. Electric trimmer was used to remove excess of the hair from different body parts. Commercial hair remover (NAIR™) was applied to the skin to remove the remaining hair. Tape-striping using lab-tape and Kimwipes™ was performed until the skin glistered. Skin samples at different body regions, ~3 cm² size, were then dissected out and rinsed with PBS 2-3 times for 5 minutes. Each skin piece was further cut into small pieces. The skin samples were fixed for another 2-3 h in 4% PFA/PBS at 4 °C, and any excess fat and hair were removed followed by three times PBS rinses at room temperature. Washing was done every 30 minutes for at least 4-5 hours with PBST (0.5% TX100) at room temperature. Primary antibodies (Chicken anti- NF200 Heavy (Aves Labs) 1:500, Rabbit anti-S100 (Abcam) 1:500, Rabbit anti-NF200 (Sigma) 1:1000, Rabbit anti-GFP (Invitrogen) 1:1000, Rabbit anti-DsRed (Clontech) 1:500, Rat anti-K8 or TROMA-1 (DSHB) 1:100) in chilled blocking solution (5% heat inactivated goat serum, 75% PBST, 20% DMSO) were applied and incubated on rocker at room temperature for 72 hrs. Tissues were rinsed in PBST for 3 times followed by a 30-minute wash for at least 4–5 hours with PBST. Tissues were then incubated in secondary antibody in blocking solution at RT for 48 hrs. PBST was used to rinse the tissues for 3 times followed by a wash every 30 minutes for at least 4-5 hours with PBST. Tissues were dehydrated (1–2 hrs for each) in serial MeOH (Methanol)/PBS dilutions (25, 50, 80,

100) rocking at RT. Tissue was kept in 1:1 MeOH:BABB (1 part Benzyl Alcohol: 2 parts Benzyl Benzoate) solution in 10 mL glass vial for 1-3 hrs rocking at room temperature.

Tissue was cleared in 100% BABB and mounted on a slide (with little BABB). Four drops of grease were put at four corners around the tissue and a cover glass was put over it with gentle pressure so that the coverslip sticks with the mounting tissue/grease.

## RNAscope in situ hybridization

Intact spinal columns and lumbar DRGs were dissected from $CO_2$ euthanized mice and rapidly frozen in OCT on a dry-ice/ethanol bath. Using a cryostat, 20 µm cryosections were collected on Superfrost Plus slides (Fisher, 22–034-979) and allowed to dry for at least 2 hours at room temperature. RNAscope in situ hybridization was performed in accordance with the manufacturer's instructions using RNAscope™ Multiplex Fluorescent Reagent Kit v2. RNAScope probes, Mm-*cFos* (316921-C3), Mm-*Slc32a1* (319191), Mm-*Slc17a6* (319171-C2), Mm-*Eyfp* (312131), Mm-*tdTomato* (317041), Mm-*iCre* (423321-C3), Mm-*Ret* (431791-C2), Mm-*Calb1* (428431-C2), Mm-*Ntrk2* (423611-C3) were purchased from ACD (Advanced Cell Diagnostics, Inc.).

## Electrophysiology

**Ex vivo skin-nerve preparation.** *Split$^{Cre}$;Advil$^{FlpO}$;Rosa$^{ReaChRf/+}$* mice of both male and female sexes were used, and most of them (31 mice) were aged 6 to 15 weeks and 4 were over 20 weeks. In one set of experiments, naïve animals were used, and in another set of experiments, animals were randomly assigned into saline and CFA groups. In saline group, each animal was injected with 10 µl saline into both hind paws. In CFA group, each animal was injected with 10 µl CFA (5 µg/10 µL) into both hind paws. 4–7 days after the injection of saline or CFA, animals were anesthetized with 5% isoflurane and then sacrificed by decapitation. The saphenous nerves with their innervated hairy skin of the hind paws or the tibial nerves with their innervated glabrous skin of the hind paws were dissected out from the animals. The skin-nerve preparation was then placed in a Sylgard Silicone-coated bottom of a 100-mm recording chamber that contained the Krebs bath solution described below. The fat, muscle and connective tissues on the nerves and the skin were carefully removed with a pair of forceps. The skin was affixed to the bottom of the chamber by tissue pins with epidermis side facing down, and the nerve bundle was affixed by a tissue anchor in the same recording chamber. The recording chamber was then mounted on the stage of the Olympus BX51WI upright microscope. The skin-nerve preparation was superfused with a normal Krebs bath solution that contained (in mM): 117 NaCl, 3.5 KCl, 2.5 CaCl$_2$, 1.2 MgCl$_2$, 1.2 NaH$_2$PO$_4$, 25 NaHCO$_3$, and 11 glucose (pH 7.3 and osmolarity 325 mOsm) and was saturated with 95% O2 and 5% CO$_2$. The Krebs bath solution in the recording chamber was maintained at 24°C during experiments recordings. To facilitate the pressure-clamped single-fiber recordings, the cutting end of the nerve bundle was briefly exposed to an enzyme solution that contained 0.1% dispase II and 0.1% collagenase in Krebs solution for 30 to 60 s, and the enzyme was then washed off by the continuous perfusion of the normal Krebs solution.

**Pressure-clamped single-fiber recordings.** The pressure-clamped single-fiber recording was performed in a manner similar to our previous studies[51]. In brief, recording electrodes were made by thin-walled borosilicate glass tubing without filament (inner diameter 1.12 mm, outer diameter 1.5 mm). They were fabricated using a P-97 Flaming/Brown Micropipette Puller and fire polished to make tip diameter at 10 to 50 µm. The recording electrode was filled with the Krebs bath solution, mounted onto an electrode holder which was connected to a high-speed pressure-clamp device (ALA Scientific Instruments, Farmingdale, NY). Under a 40x objective, individual nerve fibers in the cutting end of the whisker afferent nerve bundle were separated by a positive pressure of approximately +10 mmHg delivered from the recording electrode. The end parts of nerve fibers in the number of 3 to 30 were then aspirated into the recording electrode by a negative pressure at approximately −10 mmHg. Once the nerve ends reached approximately 10 to 30 µm in length within the recording electrode, the pressure in the recording electrode was readjusted to −5 to −1 mmHg and maintained throughout the experiment. To identify nerve fibers that are ReaChR/YFP⁺, a beam of red LED light (wave length 617 nm) was focused through the 40x objective to the skin to search the receptive field where action potential impulses could be evoked by the LED light. ReaChR can be activated by blue and orange lights (Lin et al.[52]). In searching the light sensitive receptive field, LED light were continually applied with 5-ms light pulses at the frequency of 0.1 Hz, and the intensity of the LED light was at 4.5 mW. To determine mechanical sensitivity of the light-sensitive receptive field, a mechanical indenter (Aurora scientific: 300C-I) was used to apply mechanical stimulation at the pre-identified light-sensitive receptive field. The tip size (in diameter) of the indenter was 0.3 mm for the hairy skin and 0.8 mm for the glabrous skin. The mechanical stimulation was applied under the force control module in which a ramp-and-hold stepwise force was applied to the skin. Prior to the application of the stepwise force, the tip of the indenter was lowered to the surface of the receptive field with a 1-g force and then the 1-g force was canceled to 0 so that the tip of the indenter was just in contact with the receptive field surface but without having any force applied to the receptive field. The indenter was connected to a Digidata 1550B Digitizer and the stepwise force was delivered to the indenter using the command from pClamp 11 software. The step force commanders were calibrated by applying indenter at finger tips, paw pads and other areas of plantar skin, and the actual forces applied to these skin areas were measure and used to correct the commander force steps. The actual ramp-and-hold force steps were applied from 0 mN to 5, 30, and 80 mN. The duration of the ramp (dynamic phase) was 10 ms, and the duration of the holding step (static phase) was 0.98 s. The minimal force step at which AP impulses was elicited was defined as mechanical threshold of the receptive field. Two types of mechanical responses were observed in the light-sensitive receptive field, the Aβ RA-LTMRs for which AP impulses occurred only in the dynamic phase of mechanical stimulation, and the Aβ SA-LTMRs for which the AP impulses occurred in both the dynamic and static phases of mechanical stimulation. The signals were recorded using a Multiclamp 700B amplifier and signals were sampled at 25 kHz with band path filter between 0.1 Hz and 3 kHz on AC recording mode.

In a different set of experiments, conduction velocity of the ReaChR/YFP-positive saphenous nerves were determined. In this set of experiments, ReaChR/YFP-positive nerves were visualized under a fluorescent microscope and a loose-patch recording was made at the node of Ranvier in a similar manner described in our previous studies[53]. Impulses were evoked at the peripheral end of the saphenous nerve bundle using a suction stimulation electrode. The suction stimulation electrode's tip size was approximately 0.5 mm in diameter and was fire polished. The peripheral end of the saphenous was aspirated into the suction stimulation electrode with a tight fitting by negative pressure. The negative pressure was continuously applied into the suction stimulation electrode to maintain the tight-fitting during experiments. To initiate AP impulses at the peripheral end of the saphenous nerve, monophasic square wave pulses were generated by an electronic stimulator (Master-8, A.M.P.I, Israel) and delivered via a stimulation isolator (ISO-Flex, A.M.P.I, Israel) to the suction stimulation electrode. The duration of each stimulation pulse was 50 µs and the stimulation intensity for evoking impulses were 100 to 200 µA.

## Drug-delivery

**Diphtheria Toxin (DTA) administration.** Either Diphtheria Toxin (20 µg/kg, Sigma-Aldrich, USA) or the water (vehicle) was injected (intraperitoneal) into *Split$^{Cre}$;Advil$^{FlpO}$;Tau$^{ds-DTRf/+}$;Ai9$^{tdTomatof/+}$* mice at the

age of 6-week old for 7 days. Two weeks after the last injection, some mice were sacrificed and perfused. Their tissues (spinal cord, DRGs, and skin) were collected for histological characterization to determine efficiency of DTA-induced ablation. The remaining mice were used for different behavioral tests from one month after the last DTA or vehicle injection.

## Mouse pain models

**CFA-induced inflammatory pain model.** The CFA-induced inflammatory pain model was generated as previously described[54]. Briefly, a mouse with the desired genotype was anesthetized by isoflurane, and the plantar surface of one paw was sterilized. 10 µL of CFA emulsion (Sigma, F5881) was slowly injected into the plantar skin surface (from the lower middle walking footpad at the ventral surface towards the plantar area) using a 0.5 mL insulin syringe. Injection site was hold using gentle thumb pressure to avoid leakage of CFA for ~10 s. The success of model was confirmed by measuring the paw-thickness by a digital electronic caliper (#30087-00, Fine Science Tools) (1 day post injection) and mechanical sensitivity using von Frey filaments.

**Medial Plantar Nerve Ligation (MPNL)-induced neuropathic pain model.** The MPNL model was generated as previously published[32]. Briefly, a mouse with the desired genotype was anesthetized using isoflurane. After sterilizing the paw skin, the medial surface of the ankle of one leg was incised (0.5 cm) using #11 blade to expose the medial plantar nerve. One ligation was performed with a 4−0 cat-gut suture (Ethicon). The skin was sealed. After waking up, mice were returned into the home cage.

## Behavioral assays

**von Frey Hair assay.** To test the static touch sensitivity, mice were habituated for 2 days in the behavioral room. Under Plexiglas chambers (11.5 × 4.5 × 4 cm), they were kept over a perforated wire mesh platform (Ugo Basile, Italy, Part #37450-045) which had 5 × 5 mm gaps, for 1 h before starting the experiment. Paw withdrawal threshold was assessed using the up-down method[55] as described previously[56]. Briefly, 8 calibrated and logarithmically spaced von Frey monofilaments (bending forces: 0.008, 0.02, 0.04, 0.07, 0.16, 0.4, 0.6 and 1 g; Stoelting, Wood Dale, IL) were used. These were applied tangentially to the plantar surface for ~3−4 s with enough force to cause a slight buckling of the filament. First, the middle filament (0.16 g) was applied to the hind paw. If the mouse showed a withdrawal, an incrementally lower filament was applied. If there was no response, an incrementally higher filament was applied. Trials were separated by at least 2−3 minutes to avoid sensitization and learning. A positive response was characterized as a rapid withdrawal of the paw away from the stimulus fiber within 3−4 s (Sudden withdrawal, flicking or licking of the tested paw). Testing was continued until four filaments were applied (heavier or lighter, depending on the exact filament size to which the last response occurred) after the first one that produced a withdrawal. The final value of 50% withdrawal threshold was calculated by using pseudo-log calculator in Excel which uses the following equation[55]:

$$50\% \text{ withdrawal threshold}(g) = (10^{[Xf + k\delta]})/10{,}000$$

Where $Xf$ = value (in log units) of the final *von* Frey filament used; $k$ = tabular value for the pattern of positive/negative value and $\delta$ = mean difference (in log units) between stimuli.

The observer was blind to the genotype and the drug-treatment while performing the behavioral experiments.

**Cotton swab assay.** After performing von Frey testing, on the consequent day, gentle dynamic touch-evoked sensitivity was measured as described previously[57]. Mice were placed under transparent Plexiglas chambers on an elevated wire-mesh platform floor. The floor consists of mesh-like grids that are accessible from below due to small gaps of ~5 × 5 mm. Mice were habituated 1 h daily for 2 days on this setup in the behavior room and allowed to acclimate for 1 hour before testing. A cotton swab from a cotton applicator (Puritan 25-806 1WC) was manually pulled so that it was "puffed out" to ~3X the original size. When the mouse was at rest, a constant sweeping motion from heal towards the toes was used underneath the mouse paw. Mice were recorded using high-speed imaging camera (500, 1000 fps). Their paw withdrawal response and other spontaneous behaviors were analyzed. Five trials were performed with ~5 minutes interval between each sweep. Both hind paws were used randomly. The videos were saved in an external hard drive and later analyzed for the number of hind paw withdrawals out of 5 times as a percentage (%) response for each mouse and averaged.

**Tape removal assay.** This assay was performed over glabrous skin (plantar surface) of the mice slightly modified from the[58]. Mouse was habituated in a transparent plexiglas rectangular chamber over an elevated perforated wire-mesh platform for 45 minutes. Mouse was removed from the enclosure, and a 9.5 mm diameter, red, circular adhesive Microtube Tough-Spots label (RPI 247106R) was attached to the plantar surface of the hind paw. The mouse was returned to the chamber immediately for video recording. The behavior of mice was recorded using a HD web camera (C922 Pro HD Stream Webcam). The time that each mouse took in removing the tape was measured. 5 minute was the cutoff time for the experiment.

**Texture-preference assay.** A chamber with two compartments separated by a smaller mid-compartment was designed. The two big chambers' floors were covered with loop (Soft side) or hook (Rough side) of Velcro tape, respectively. Also, each chambers had distinctive visual cue patterns (white and black stripes vs. white checkered duct tapes). Middle chamber separating these two chambers was completely black, and its floor was made to the level of the two other chambers using plane thick white cardboard. Experiment was done in dim light. Daily for 3 days, each mouse was given restricted habituation of 20 minutes in each chamber. On 4th day, each mouse was allowed to move freely in all chambers for 20 minutes and videotaped from the upside. These videos were analyzed using AnyMaze software for quantifying the total time spent in each chamber.

**Plantar-pinch assay.** Mice were habituated in a transparent cylindrical glass chamber (Diameter 10 cm) for 30 minutes. An alligator clip (Amazon, "Generic Micro Steel Toothless Alligator Test Clips 5AMP"), which produces 340 g force, was applied to the ventral skin surface between the footpads. The animals were returned into the chamber, and their behaviors were video recorded using a high-speed imaging camera (80−200 fps) for 60 s. Behaviors, such as licking, wiping, shaking and scratching episodes, were quantified. Those animals with the clip somehow removed before 60 s cutoff were not considered in the study.

**Hargreaves plantar assay.** Mice were placed in opaque rectangular Plexiglas chambers over the glass surface of the Hargreaves apparatus (UCSD Instruments, San Diego, CA, USA), and allowed to acclimate for ~45 minutes. Mice urine or feces, if any, were removed and the hind paw and the glass surface were cleaned and dried using wipes immediately. A thermostat was used to assess the constant temperature of the glass plate (~22 °C ± 1). The heat stimulus was applied from a bulb beneath the glass, to the middle of the plantar surface of the hind paw. Either a brisk paw withdrawal (flick) or licking in response to the thermal stimuli was considered as a positive response. Cutoff time was 20 s. Three trials with a gap of 10 minutes were taken and averaged as the paw withdrawal latency (in second).

**Open field activity assay**. Mice were habituated for two days in the behavior room in their cage for 45 minutes in the dim light condition. On the third day, mouse was directly kept in the center of the arena and the behavior was recorded using a webcam from the upside. The experimenter was not present in the room while recording the behavior to avoid observer-related behavioral changes. The total recording time was 20 minutes. Before starting experiment in the next mouse, the surface of arena was cleaned using 40% alcohol and allowed to dry for 10 minutes. The videos were analyzed using AnyMaze software to measure the time spent in the Central and peripheral zone as well as the overall distance traveled.

**Static hotplate assay**. For the static hot plate test, mice were placed on top of a hot plate (IITC, Life Science), preset to $50 \pm 0.5\,°C$ covered by a transparent plexiglas chamber. The behavior of the mouse was recorded using a high-speed imaging camera (200 fps). Videos were analyzed, and the latency to lick or flick the hind paw or jumping was quantified. Three trials at intervals of at least 15 min were taken and the average score for each mouse was obtained. To avoid tissue injury of the mice, a cutoff of 30 s was set.

**Incremental hot plate assay**. The incremental hot plate test was carried out on the same apparatus using different settings, with at least 24 h rest from the static hotplate assay. The initial temperature was set to $28\,°C$ and increased by $6\,°C/min$ towards a final temperature of $55\,°C$[59]. The temperature when the first hind paw lick occurred was recorded. If no hind paw lick was observed, the test was terminated at $55\,°C$. Three trials at intervals of at-least 15 min were taken and the average temperature for each mouse was obtained.

**Dry-ice assay**. Mice were placed in Plexiglas chambers on a 2.5 mm thick elevated glass plate and allowed to habituate for at least 45 min. When the mouse was completely at rest, a dry ice pellet (1 cm diameter) was applied to the lower glass surface underneath the hind paw of the animal, and the withdrawal (flick, lick, or both) latency was measured using a stopwatch. Each hind paw was tested randomly for three times with at least 15 min interval in between two consecutive trials. To avoid frost-induced tissue injury, the cutoff latency was set to 10 s.

**Tape response assay**. This assay was performed over hairy (back) skin with slight modifications[43]. Mice were habituated in a transparent Plexiglas rectangular chamber over an elevated perforated wire-mesh platform for 45 minutes. They were removed, and a small piece of laboratory tape (~3 cm × 1 cm) was placed gently on the bottom center of the mouse's back. Mice were video recorded for a duration of 5 min by a web or high-speed camera. The total number of scratching bouts, wipes and other behaviors in response to the tape were quantified.

**Nape-pinch assay**. Mice were anesthetized using isoflurane, and the nape of the neck (~2 cm² area) was shaved using an electric trimmer. After 2–3 days, the mice were habituated for 15 minutes in a glass chamber of ~10 cm diameter. An alligator clip producing 340 g force was applied to the shaved nape skin fold. The animal was placed back into the chamber and video recorded using high-speed camera (80–200 fps) for 60 s. The videos were analyzed to quantify the scratch bouts, headshakes, bilateral wipes, and attending duration/episodes.

**Tail-immersion assay**. A mouse to be tested was restrained in a plastic 50 mL screw capped conical centrifuge tubes with several holes in the tube wall so that mouse can breathe normally. A 0.5 cm² opening was cut in the cap to allow the tail access to the water bath. Mice were habituated in the tube for 30 minutes in 2 days. On 3rd day, by holding the tube horizontal, the distal part of the tail (~4 cm) was submerged in the temperature-controlled water bath. Sudden tail-flick was used as a

response sign. 3 trials were performed with an interval of 15 minutes. The tail-flick latencies of the three trials were recorded using a stopwatch and averaged for quantification. Cutoff time was 10 s. The test was performed at a gap of 24 h for 3 temperatures (48, 50 and 55 °C).

**Peripheral optogenetics, high-speed imaging, and pain score**. To study several sub-second behavioral phenotypes of *Split^Cre*-*Aβ ReaChR* mice after peripheral light-stimulation of different body parts, we used a high-speed camera (FASTCAM UX100 800K-M-4GB - Monochrome 800 K with 4 GB memory) with an attached zoom lens (Nikon Zoom Wide Angle Tele-photo 24–85 mm f2.8) on a tripod as previously described[16,31,56]. Behaviors were recorded at 100–1000 frames per second (fps) with different resolutions as per the demand of the behavior test. A far red-shifted LED light that mice cannot detect was used to help the video quality. Mice were acclimated on an elevated perforated wire-mesh platform daily for 1 h in transparent rectangular Plexiglas chambers either for 2 (Plantar stimulation) or 3 days (after shaving the nape of the neck and back). On the day of the experiment, they were habituated for 45 min before peripheral 473 nm blue laser (Shanghai Laser and Optics Century, BL473T8-150FC/ADR-800A) stimulation. When mouse was still, a blue laser (5–20 mW, 10 Hz square wave, waveform generator (hp Hewlett Packard 15 MHZ Function Waveform Generator, 33210A)), was shined upon the plantar surface below the wire-mesh space (the distance between paw and the laser cord-outlet tip is ~2 mm). Laser intensity was measured using a Digital Optical Power Meter with a 9.5 mm aperture (ThorLabs, PM100A). Each mouse was tested for 5 trials with an inter-trial interval of 5 min. The percentage of trials showing paw withdrawal response, such as the paw flutter, flick, lick or the brisk withdrawal of the paw, was quantified. Only data from fully habituated mice (no movement before light stimuli in high-speed imaging videos) were quantified. The paw-withdrawal latency (s) in response to the plantar optogenetic stimulation was also quantified and averaged from 5 trials.

For optical stimuli of the tail, when the mouse was not grooming and its tail was at complete rest, the mid of the tail was stimulated by shining a blue laser from ~2 mm distance below the wire-mesh. Each mouse was tested for 5 trials with an inter-trial interval of 5 min. The latency to flick the tail and the percentage of trails showing withdrawal response were quantified and averaged from 5 trials.

For optical stimuli of the nape and back, mice were shaved at these regions using an electric trimmer and then habituated for 3 days under transparent plexiglas chambers. Nape or back of these mice were stimulated from upside of the chamber by a blue laser when they were still. Any sudden body-movement/shaking or avoidance behaviors of the mice was considered as a positive response. Each mouse was tested for 5 trials with an inter-trial interval of 5 min. The latency to flick the tail and the percentage of trails showing withdrawal response were quantified and averaged from 5 trials.

The pain score was quantified as previously described[16]. Briefly, four individual behavior features: orbital tightening, hind paw shake, hind paw guarding, and jumping were considered as "pain" related behaviors, and 1 score was given for each behavior shown (0 minimum and 4 maximum score) for a testing trial. The final pain score was averaged from all five trials.

**Conditioned place preference assay**. A chamber with two compartments separated by a smaller central compartment was designed. The two big chambers' walls had distinctive visual cue patterns (white and black stripes vs. white checkered duct tapes). The apparatus was placed over the elevated wire-mesh platform. Each mouse was allowed 20 minutes to explore the three-chambered apparatus with no optogenetic stimulation for 3 days. The mouse was excluded from the study if it showed more than 60% preference to a particular chamber at this stage. From days 4–6, whenever mouse entered a specific chamber

(White strips pattern on the wall), a blue laser (5 mW, 10 Hz) was shined on the plantar surface of the treated hind paw (either saline- or CFA-treated) from below the wire mesh platform till the mouse leaves this chamber and reach to the other chamber. To avoid any prospective optic cable and hand movement-related non-specific response, the optic cable (held in the hand of the experimenter) was also moved under the platform with the movement of the mouse but the light was only shined when the mouse entered the stimulation chamber. This was repeated for 20 minutes from days 4–6. On day 7, we tested these mice for place preference where each mouse was allowed 20 minutes to freely move about the three-chambered apparatus without blue laser stimulation. The activity was recorded using a webcam from the upside and scored later using AnyMaze software. Percent post-stimulation change was calculated as percent time in blue light chamber after training minus percent time in blue light chamber before training.

### Procedures for spinal cord optogenetics

**Spinal light cannula implantation.** A light cannula (Ceramic, 1.25 mm diameter, 200 μm optic core, and length ~0.25 mm, Thor Labs) was implanted in T11 region of the mouse spinal cord as described previously[60]. Briefly, under isoflurane anesthesia, the hairs of the back region of a mouse were removed, and the mouse was placed in a stereotax (Model 940, KOPF instruments, Tujunga, CA, USA). After sterilization of the surgical site, a local anesthetic was administered prior to the incision. For implantation at ~T11 region, a 1 cm long incision was made starting caudal of the peak of the dorsal hump, extending approximately 0.5 cm rostral and 0.5 cm caudal from the initial incision site. White tendons were cut from both sides of spinal column and the vertebral column was exposed by clearing tissue from the transverse processes without damaging spinal cord and nerves. T11 vertebra was fixed from both sides using the spinal adapters (#51690, Stoelting Co., Wood Dale, IL). Surface connective tissue was removed from the T11 vertebra and the adjacent rostral and caudal vertebrae by gentle scrubbing using the tip of sterilized cotton swabs. With a fine tipped burr drill (0.5 mm in diameter), the bone and dura mater were punctured. A light cannula was prepared to implant. The hole surface was dried using the cotton swabs. Using the other side tip of the cotton applicator, a very little amount of glue (Krazy®Glue, Part #963257) was applied around burr-hole before lowering the cannula into place. After lowering the cannula through the hole, and once the cannula attaches to the spinal cord dorsal surface firmly, dental cement was applied around the outside of the vertebra to stabilize the cannula. Spinal fixation bars were removed after drying of the cement. The skin was sutured, followed by administration of systematic analgesics. Mouse was put on a warm pad for recovery. After surgery, each mouse was housed individually to avoid accidental removal of the light-cannula by a cage mate. Implanted mice were used for behavior assays 2 weeks after implantation.

**Spinal optogenetic stimuli.** Two weeks after the spinal light-cannula implantation into *Split^Cre-Aβ ReaChR* mice, the light cannula was attached to a rotating fiber cannula connected to the blue laser. These mice were habituated on an elevated perforated wire-mesh platform daily for 1 h in a transparent circular Plexiglas chamber (7 cm diameter and 30 cm height) for 3 days. On 4th day, in the similar settings, after 45 min habituation, mice were stimulated by a blue laser of lower (0.5 mW) or higher intensity (10 mW) (an interval of 24 h between the two laser tests). The stimulation protocol ('n' mW, 10 Hz, square wave, 30 s on, 1 min off) was performed 10 times. VFH testing to check the mechanical sensitivity of the plantar surface was performed at 15 minutes before stimulation (baseline) and different time points (5, 30, 90 and 120 min) after stimulation. Mouse behaviors were recorded at 30 fps using a normal web camera

(Logitech) to detect any spontaneous behavioral phenotype (Licking, scratching, etc.). Some mice were sacrificed for spinal c-Fos immunostaining in 90 minutes after the last stimulation. Control mice were sacrificed without light stimuli.

**Data collection, quantification, software, and data presentation.** Both male and female mice were used for all experiments, and they were separately analyzed initially. Since no sex difference was evident, in later experiments, the data from both sexes were pooled. Electrophysiological data were analyzed using Clampfit 10 (Molecular Devices, Sunnyvale, CA, USA). Data were collected from 17 male and 14 female animals and were aggregated for data analysis. To confirm that impulses evoked by LED light and mechanical indenter are generated from the same fiber, the amplitudes and shapes of the impulses evoked by both LED light and mechanical indenter at the same receptive field were compared, and the data were included only when mechanically evoked impulses matched the light-evoked impulses. Conduction velocity was calculated by the distance between stimulation site and recording site divided by the time latency for eliciting an impulse following electrical stimulation. Histological images were taken using a Leica SP5II confocal microscope (fluorescent). Image processing and scale bar addition were performed in Fiji-ImageJ. Cell-number counting was performed in FIJI software (NIH). High-speed imaging videos were collected on FASTCAM Viewer 4 (Photron). Graphs were originally created using GraphPad Prism Version 9.4.1 and further modified in Adobe Illustrator. Cartoons in the figures were created by authors using either BioRender.com or Adobe Illustrator. All figures were generated using Adobe Illustrator.

**Statistics and reproducibility.** All data shown in column and line graphs represent Mean ± SEM, unless otherwise mentioned. Significance levels are indicated as $*p < 0.05$; $**p < 0.01$; $***p < 0.001$; $****p < 0.0001$. Sample sizes and statistical methods are mentioned in respective figure legends. Parametric data were analyzed using a Student's $t$-test, one- and two-way ANOVA. (a mixed-effects model instead of a two-way ANOVA if random value missing) and Tukey's, Holm-Šídák's or Bonferroni's post hoc tests when appropriate, as indicated in figure legend. For non-parametric data, the two tailed Mann-Whitney test for two-sample comparisons or the Kruskal–Wallis test for multiple comparisons was used.

The DRG, spinal cord, and skin sections or whole mount pieces were randomly chosen for IHC and RNAScope experiments. Animals of the correct genotype were randomly selected to give either vehicle or DTA, and saline or CFA. The conduction velocity of ReaChR/YFP+ fibers were randomly recorded. The experimenter was blind to genotypes and treatments of the mice for performing most of the behavioral experiments. Sample size was determined based on previous publications with similar models and experiments. To ensure replicability, results were derived from at least three independent experiments including the micrographs generated in this study. The number of replications for each experiment was included in the figure legends. No data were excluded from the analyses.

In vitro electrophysiological recordings were not performed in a manner blinded to genotype, as only triple mice expressing ReaChR2 were recorded, or treatment, due to the easily visible inflammatory phenotype of the CFA treated hind-paw. The experimenter was not blind to the chronic pain models (CFA-induced inflammatory pain or MPNL neuropathic pain) due to the easily visible inflammatory phenotype of the treated hind-paw and the paw guarding posture of the affected mice.

### Reporting summary
Further information on research design is available in the Nature Portfolio Reporting Summary linked to this article.

## Data availability

All data needed to evaluate the conclusions in the paper are available in the paper and the supplementary information. The raw data for all main and Supplementary Figs. are available in Source Data files, accompanying this paper. For information and requests related to electrophysiological recordings of this manuscript, please contact Dr. Jianguo Gu (jianguogu@uabmc.edu). For inquiries related to other results and experiments in this manuscript, please contact Dr. Wenqin Luo (luow@pennmedicine.upenn.edu). Source data are provided with this paper.

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

## Acknowledgements

We thank Drs. Sotatsu Tonomura and Ryan Vaden from the Gu lab for their help in electrophysiological experiments. We thank Dr. David Ginty for generously sharing *Split^Cre* and *Advillin^FlpO* mouse lines. We thank Dr. Martyn Goulding for generously sharing the *Tau^ds-DTRf/f*;*Ai9^tdTomatof/f* mouse line. We thank the previous and current lab members in Gu, Luo, and Ma labs for their help and insightful suggestions. We used BioRender.com to generate illustrations in this manuscript. This work is supported by NIH grants NS109059 and DE018661 to J.G.G., and NS083702 to W.L.

## Author contributions

Conceptualization, M.G., J.G.G., and W.L.; Methodology, M.G., A.Y., A.I.Y., J.G.G., and W.L.; Investigation, M.G., A.Y., A.I.Y., P.D., K.K., Q.W., and J.L.; Writing-Original Draft, M.G., and W.L.; Writing – Review & Editing, M.G., P.D., K.K. H.Y., M.M., J.G.G., and W.L; Funding Acquisition, J.G.G. and W.L.; Resources, M.M., J.G.G., and W.L.; Supervision, M.M., J.G.G., and W.L.

## Competing interests

The authors declare no competing interests.
