## [Peer Review File · Nature Communications]

REVIEWER COMMENTS

Reviewer #1 (Remarks to the Author):

Gautam et al.,

Touch is mediated by a combination of low and high threshold mechanoreceptors (LTMRs and HTMRs). There are several types of LTMRs and HTMRs, with differing tuning, electrical properties, and anatomy. Under normal conditions, fast conducting, highly myelinated sensory neurons, known as the A-beta-LTMRs, mediate highly sensitive “discriminative” touch and vibration detection; sensations that tend to be considered innocuous. However, when tissues become damaged, infected and/or inflamed, the same types of gentle mechanical stimuli instead can produce pain. How this transformation (mechanical allodynia) occurs continues to be debated and is the topic by this manuscript by Gatam et al.

The authors set out to examine the paradoxical hypotheses that Ab-LTMRs may suppress pain yet also produce allodynia. To study the role of Ab-LTMRs, Gatam et al use genetic strategies to ablate, activate or silence these neurons under basal and pathological conditions. They also develop a unique optogenetic approach to compare focal with more global stimulation. They conclude that their experiments provide support for the model that widespread Ab-LTMR activity suppresses pain whereas local activation can result mechanical allodynia under certain conditions.

The main strength of the study is conceptual. The question being studied is important and the approach is potentially powerful and creative. The main limitations of the study are the genetic strategies need to be better characterized and appear limited in terms of specificity and completeness, shortcomings that would need addressing to weigh the validity and robustness of the evidence.

Specific points:

Fig. 1, the Split-Cre mouse has minimal characterization.

- Although used in prior literature, the Split-Cre mouse has never been clearly explained. How was the transgenic mouse made? What gene or genes are being target? There doesn't seem to be a gene called “Split” expressed in LTMRs (or DRGs for that matter). Is “Split” then referring to a “split Cre” approach?

- There has been extensive transcriptomic profiling of somatosensory neurons. There are genes that in combination can clearly distinguish between the select types of LTMRs. The antibodies used here are not

sufficiently specific. The Split-Cre line needs to be more precisely characterized using multiplexed in situ hybridization.

- There should be an accounting of which neurons express Cre and how faithful the recombination is.

- The IHC images in Fig. 1 have a lot of background GFP labeling that make it unclear which cells are labeled and the quality of staining is quite variable in each panel.

- What percentage of A-beta neurons are being labeled by this strategy? Which Ab-LTMRs are these (field-LTMRs, RA-LTMRs, SA-LTMRs or some combination of these)? How many of the LTMRs express Cre or the ROSA reporter (eg is it 80% of RA-LTMRs and 20% SA-LTMRs or maybe some Ad-LTMRs are also labeled?).

Fig. 2, the DTA mice have only ~50% ablation of the Split-Cre neuronal population.

- It is essential to know which neurons exactly are being ablated, and which ones are left over?

- Nerve endings are reduced in number in plantar skin, but what about lanceolate endings in hairy skin?

- Given the mechanism by which DTA kills neurons, it is difficult to understand the conclusion the axon terminals are affected but the cell bodies are not.

Fig. 3:

- The texture assay interpretation would benefit from comparing vehicle to DTA mice in each texture (rather than comparing textures within mouse groups).

- The von Frey data would be easier to view and interpret if the force (g) is plotted on a log scale.

- Prior literature has shown that ablation of Trkb+ LTMRs that includes the Meissner endings (Neubarth et al Science 2020) eliminates mechanical allodynia (Peng et al Science 2017 and Dhandapani et al Nat Comms 2018). However, in this paper, since 50% of a subset of A-beta fibers are ablated, it is very difficult to interpret the behavioral data in the DTA mice due to the undefined population of cells being ablated.

Fig. 4, the results from optogenetic stimulation of Split-Cre are entirely expected from von Frey stimulation.

Fig. 5:

- There needs to be more neurons recorded to make a clear conclusion.

- The observation that, in chronic inflammation, rapidly adapting cells become slow adapting does not make physiological sense, because RA and SA properties are thought to be determined by static factors like genetic and end organ identity (i.e. Merkel/Ruffini endings are SA while Meissner/Pacinian endings are RA).

- Panel E has an inconsistency between the graph and the legend n numbers. Also, the line graphs should be made more visually distinct to distinguish between saline and CFA.

Fig. 6, the magnitude of response between B and F is the same post-optogenetic stimulation, suggesting that the stimulus is driving the effect regardless of the CFA treatment.

Reviewer #2 (Remarks to the Author):

The role of large-diameter light-touch mechanoreceptor neurons in mechanical hyperalgesia remains unknown and controversial within the field. The Gate Control Theory, in which activation of large-diameter, light touch sensory neurons inhibits the transmission of small-diameter nociceptors within the spinal cord preventing pain sensation, was proposed over half a century to explain why light-touch alleviated pain and led to the development of clinical interventions such as Transcutaneous Electrical Nerve Stimulation. However, several studies contradict this model and suggest that AB-LTMRs may mediate mechanical hyperalgesia of chronic pain. To reconcile these differences, Gautam and colleagues used refined genetic strategies coupled with electrophysiology, high-speed behavioral analysis, and optogenetics to examine the role of AB-LTMRs in the naïve and chronic injury state. Using an intersectional genetic strategy to label a large subset of AB-LTMRs while minimizing off-target recombination in non-neuronal glial cells (SplitCre; AdvilFlpO), Gautam, Yamada, Yamada et al found that genetic ablation of SplitCre- β -LTMRs increased mechanical pain but did not affect thermosensation in acute and chronic inflammatory pain conditions, suggesting that β -LTMRs play a modality-specific role

in gating mechanical pain transmission. Conversely, optogenetic activation of these same neurons elicited higher pain scores following tissue inflammation in comparison to the naïve state, in which activation of the same subset led to behavioral responses with low pain scores. The authors suspected that the differing results of ablation vs activation of SplitCre-A β -LTMRs arises because of distinctive local and global roles of the SplitCre-A β -LTMRs in transmitting and alleviating mechanical hyperalgesia of chronic pain, respectively. In support of this model, the authors observed that broad activation of these same neurons activated at the level of the dorsal column alleviated mechanical hypersensitivity associated with chronic inflammation. Overall, the refined genetic approach taken in this study provides novel insights into the function of A β -LTMRs in chronic pain. Overall, this is a compelling study that will be of interest to researchers in the field. However, there are some concerns or suggestions discussed below that would help strengthen the study.

Major:

-The study would benefit from further histological analysis of the SplitCre; AdvilFlpO line. Specifically, it seems that there is some amount of innervation by SplitCre neurons in layers I/II of the spinal cord (Figure 1F-G), although the authors state in the text that “sections showed that GFP+ central terminals did not innervate layers I/II”. Additionally, there are a handful of GFP+/NFH- terminals in the skin (Figure 1H) and an amount of double labeling with CGRP. It would be helpful for the authors to better characterize the GFP+/NF200- neurons in glabrous and hairy skin—are these CGRP+ HTMRs? Are any NFH- lanceolate endings (C-LTMRs or Ad-LTMRs) labeled by SplitCre? It would also be helpful to show GFP/NF200/Tuj1 staining of lanceolate endings in hairy skin to appreciate whether any other LTMR population is labeled. It is difficult to really appreciate the overlap between NFH and GFP in lanceolate endings from the image shown in S1G. Understanding whether CGRP+ neurons in the skin are labeled by SplitCre would be helpful in interpreting why optogenetic activation of this population (at 10 and 20 mW) leads to nocifensive responses in the naïve state. Also for Fig 1A, it would be helpful to show CGRP staining in red so that it would be easier to appreciate any potential overlap between GFP/CGRP.

-The authors claim that “in chronic inflammation, the electrophysiological properties of many RA AB-LTMRs became SA-like, which could be one of the underlying mechanisms for triggering stronger local activity and inducing mechanical hyperalgesia in the CFA model”, does not seem supported by the data for the following reasons:

1. Following CFA, the number of impulses for AB RA-LTMRs increased from 1 to 2—this is far from the value reached by AB SA-LTMRs.
2. The reduction in RA and increase in SA responses following CFA may be due to the decrease in GFP+ fibers innervating Meissner corpuscles, which may render these neurons unresponsive to mechanical stimulation. As such, there may just be fewer AB RA-LTMRs following CFA.

-Along these lines, the increase the authors describe for the SA neurons is based on a recording of 1 SA neuron in the control condition. I think the potential increase in SA firing is interesting and therefore it

would be worth trying to recording from 2-3 more SA neurons in the control condition to enable statistical comparisons.

-The authors show activation of the dorsal column alleviates pain (paw withdrawal threshold, Figure 6F) but this behavioral effect lasts 5 to 30 minutes after optogenetic stimulation. The authors should comment on what they think this is happening for this prolonged period of time.

-More statistical detail is needed (the ANOVA tests should report the p value, F statistic, and degrees of freedom in addition to the post-hoc test p value)- regarding the test in 4I would be nice. Was an ANOVA done here? If so, why do the authors think the littermate CFA animals are not different from the CFA animals with ReaChR activation in this conditioned place assay? Do the CFA animals at baseline prefer one side of the chamber?

Minor:

-The description of the genetic strategy in lines 194-196 is confusing and should be clarified. Currently the way it is written makes it sound like Tau-DTR is just Cre-dependent.

-The results of the texture preference assay (Fig. 3E) suggest the DTR-treated mice have a heightened sensitivity to floor texture, in comparison to the control animals. This is the opposite of the authors interpretation (265-267). Could the author please elaborate on the discrepancy in light touch sensitivity between the gentle touch assays and this texture preference assay?

-It would be helpful for the authors to compare the pain score of SplitCre activation in the naïve and chronic pain state to optogenetic activation of a nociceptive population. Is a pain score below 1 painful? It is unclear why optogenetic activation of SplitCre neurons with higher intensity light would lead to nocifensive behaviors (Fig 4D). Could the authors comment on this in the paper?

-The legend for Figure 4I describes more than the one figure panel and should be edited.

-It is challenging to appreciate the location of the Merkel cell/SA1 complex and the Meissner corpuscles in the confocal images in S5D-G either because of low magnification or unideal sectioning plane. Perhaps these examples can be swapped with more representative examples that clearly show examples of Merkel cell complex and Meissner corpuscles.

-In Figure 5E the Saline SA category says there were 2 neurons recorded but in the text and in 5D and 5G, the authors describe only 1.

Reviewer #3 (Remarks to the Author):

This is a well conducted study by Luo's group in collaboration with Gu's group. The role of low-threshold A β -low threshold mechanoreceptors (LTMRs) in pain modulation under the physiological and pathological conditions are not fully understood. In this study, authors combined intersectional mouse genetics, opto-tagged electrophysiology recordings, and high-speed imaging behavioral assays to investigate the contribution of A β -LTMRs to pain transduction and transmission. They showed both global ablation and local activation of A β -LTMRs promoted mechanical hyperalgesia. They also showed that global activation of these afferent fibers via dorsal column activation alleviated inflammatory pain via activation of inhibitory neurons in the spinal cord dorsal horn. It is very useful to generate triple heterozygous SplitCre;AdvilFlpO;RosaReaChRf/+ (SplitCre -A β ReaChR) mice as a research tool. The electrophysiological characterization is also very impressive. It is interesting to show that in chronic inflammation the electrophysiological properties of many RA A β -LTMRs became SA-like. Overall, the findings are interesting and provide new mechanistic insights into A β -LTMR mediated pain control.

Major:

1. Fig. 1A and Fig. 2B. There is some inconsistency. It seems many more neurons are labeled in Fig. 2B. than in Fig. 1A.
2. Line 213: only a 45.3% decrease of tdTomato+ DRG neurons was found after DTA treatment. Do you think the expression pattern of your marker may change after inflammation or nerve injury?
3. It appears the ablation was only conducted in naïve animals. It is also important to know if A β -LTMR ablation would also affect mechanical pain after CFA or nerve injury.
4. Figure 4A-F: It appears the control group in animals with CFA or nerve injury (without stimulation or with sham stimulation) is missing.
5. Figure 6D: Is c-Fos expression shown from the lateral or medial spinal cord? A low magnification followed by high magnification will be helpful.

Other comments

I am curious if you also examined mechanical itch? A β -LTMRs have been implicated in regulating mechanical itch (PMID: 31324538).

Line 303: The ablated mice displayed significantly increased scratching behavior in response to the tape. Scratching may be related to itch?

Line 350: same stimuli evoked stronger responses of SplitCre-A β -LTMRs in CFA-induced chronic inflammatory pain condition. I wonder if this increase also occurs in the contralateral side. Bilateral mechanical allodynia was reported in the CFA model.

Reviewer Comments:

Reviewer #1 (Remarks to the Author):

Gautam et al.,

Touch is mediated by a combination of low and high threshold mechanoreceptors (LTMRs and HTMRs). There are several types of LTMRs and HTMRs, with differing tuning, electrical properties, and anatomy. Under normal conditions, fast conducting, highly myelinated sensory neurons, known as the A-beta-LTMRs, mediate highly sensitive “discriminative” touch and vibration detection; sensations that tend to be considered innocuous. However, when tissues become damaged, infected and/or inflamed, the same types of gentle mechanical stimuli instead can produce pain. How this transformation (mechanical allodynia) occurs continues to be debated and is the topic by this manuscript by Gatam et al.

The authors set out to examine the paradoxical hypotheses that Ab-LTMRs may suppress pain yet also produce allodynia. To study the role of Ab-LTMRs, Gatam et al use genetic strategies to ablate, activate or silence these neurons under basal and pathological conditions. They also develop a unique optogenetic approach to compare focal with more global stimulation. They conclude that their experiments provide support for the model that widespread Ab-LTMR activity suppresses pain whereas local activation can result mechanical allodynia under certain conditions.

The main strength of the study is conceptual. The question being studied is important and the approach is potentially powerful and creative. The main limitations of the study are the genetic strategies need to be better characterized and appear limited in terms of specificity and completeness, shortcomings that would need addressing to weigh the validity and robustness of the evidence.

Specific points:

(1) Fig. 1, the Split-Cre mouse has minimal characterization.

- Although used in prior literature, the Split-Cre mouse has never been clearly explained. How was the transgenic mouse made? What gene or genes are being target? There doesn't seem to be a gene called “Split” expressed in LTMRs (or DRGs for that matter). Is “Split” then referring to a “split Cre” approach?

- There should be an accounting of which neurons express Cre and how faithful the recombination is.

Answer: We appreciate these points raised by the reviewer. We had illustrations in the supplementary Figure 1 (S1A) to explain this transgenic mouse line. In the revised manuscript, we have added more text in the **pages 6-7** to explain this mouse line. **“The *Split*^{Cre} mouse line was generated by GENSAT, using two bacterial artificial chromosomes (BACs) containing the *Abhd3* and *Ntng2* genes. The two halves of iCre (19-59 and 60-343) were fused with the constitutively active coiled-coil interaction domain of the yeast transcription factor GCN4 and were inserted at the start codon of the *Abhd3* and *Ntng2* genes (Supplementary Fig. 1A). With this design, the iCre recombines in cells where both *Abhd3* and *Ntng2* transgenes are expressed.”**

We also provide additional results to characterize this line during the revision (new Supplementary Fig. 1C-D). We conducted double RNASCOPE *in situ* of *iCre* (its probes recognize the C terminal containing the majority of *iCre* cDNA, 849bp) and *Eyfp*. Please note that the *iCre* N terminal cDNA fragment is short (120bp) and thus not be visible by this approach. Almost all (96%) double positive DRG neurons are *EYFP*⁺, but only ~50% of double positive DRG neurons are *iCre* (C terminal) positive. These results suggest that around half of DRG neurons containing the C terminal *iCre* have co-expression of the N terminal of *iCre*, as indicated by resulting recombination (expression of *EYFP*). This pattern demonstrates the intersectional effects of the *iCre* strategy.

Since *EYFP*⁺ neurons indicate the true recombination activity of *iCre*, it makes sense to characterize overlapping expression patterns of molecular markers in *EYFP*⁺ neurons (see below).

We modified text (page 7) and supplementary figure 1 accordingly to incorporate these new results.

- There has been extensive transcriptomic profiling of somatosensory neurons. There are genes that in combination can clearly distinguish between the select types of LTMRs. The antibodies used here are not sufficiently specific. The *Split-Cre* line needs to be more precisely characterized using multiplexed *in situ* hybridization.

Answer: We agreed with the reviewer and conducted additional multiplexed *in situ* hybridization in this revision.

To further characterize the *Split^{Cre}* mouse line, we performed fluorescent RNAScope *in situ* hybridization using lumbar DRG sections of *Split^{Cre}Aβ ReaChR* mice with following markers¹:

- i. *Ret* (For Aβ field LTMRs): Fig. 1A
- ii. *Calb1* (For Aβ RA LTMRs): Fig. 1B
- iii. *Ntrk2* (Aβ and Aδ RA LTMRs): Fig. 1C. Please note that *Ntrk2* is highly expressed in a subset of DRG neurons as well as satellite glia cells surrounding all DRG neurons (enlarged image). We quantified double positive neurons by comparing both channels side by side to exclude false double positive ones (*Ntrk2* signals from the satellite glia cells).

Currently, there is no specific exclusive marker for Aβ SA LTMRs.

Quantification is shown in Fig. 1D, which suggests that *Eyfp* is preferentially expressed in *Ret*⁺ and *Calb*⁺ DRG neurons but to a less extent in *Ntrk2*⁺ neurons.

We modified text (page 8), Fig. 1, and legend accordingly to incorporate these new results.

- The IHC images in Fig. 1 have a lot of background GFP labeling that make it unclear which cells are labeled and the quality of staining is quite variable in each panel.

Answer: We thank reviewer for this comment and have replaced them with better quality images in the revised manuscript. In addition, we put the new RNAScope *in situ* hybridization results in the Fig. 1 and moved the IHC images to the Supplementary Fig. 1 E-I.

-What percentage of A-beta neurons are being labeled by this strategy? Which AbLTMRs are these (field-LTMRs, RA-LTMRs, SA-LTMRs or some combination of these)? How many of the LTMRs express Cre or the ROSA reporter (eg is it 80% of RA-LTMRs and 20% SA-LTMRs or maybe some Ad-LTMRs are also labeled?).

Answer: Our optotagged recording results (Fig. 1L) showed that all randomly recorded cutaneous ReaChR2+ fibers have conduction velocity in the A β range (100% are A β fibers). By performing multiplex ISH and RNAScope *in situ* hybridization using different markers for A β RA LTMRs (*Ret* and *Calb1*) with DRG sections, we found that *Split^{Cre}* preferentially (>80%) recombined in A β RA LTMRs (Figures 1A, B&D and Supplementary Fig S1). ~40% EFYP+ neurons are *Ntrk2⁺* (Figure 1C&D), which marks both A δ LTMRs² and a subtype of Meissner corpuscle innervating A β RA LTMRs³. Though there is no specific marker for A β SA LTMRs, given that almost all EFYP+ are NF200+ and that EYFP+ show high percentage of overlap with A β RA LTMR markers (>80%), we estimate that the amount of A β SA LTMRs labeled by this line is 20% or less. These cell body molecular marker characterization results are consistent with nerve terminal staining in the skin (Fig. 1G-I, Supplementary Figures S1J-T, and U-W) and with the optotagged recording results of the glabrous and hairy skin (Fig. 1Q). Though our results can't exclude genetic labeling of Ad-LTMRs by *Split^{Cre}* line, they must make only a very small percentage of cutaneous afferents (even less than A β SA LTMRs in the glabrous skin (~10%)), so they were not picked up by random recordings of Split-ReachR2 mice). The 40% overlap with *Ntrk2* may indicate *Split^{Cre}* recombination in some *Ntrk2⁺* Meissner corpuscle innervating A β -RA LTMRs or in non-cutaneous Ad-LTMRs. We also added quantification of the percentage of touch domes innervated by EYFP+ afferents in the hairy skin (~10% of the total touch domes were innervated by EYFP+ fibers, new supplementary figure 1W).

We modified text, Fig 1, FigS1, legends, and discussion accordingly to indicate the new results.

(2) Fig. 2, the DTA mice have only ~50% ablation of the *Split-Cre* neuronal population.

- It is essential to know which neurons exactly are being ablated, and which ones are left over?

Answer: We appreciate this important point raised by the reviewer. Previously, we used whole mount DRGs for quantifying tdT+ DRG neurons. However, there were tdTom signals generated by apoptotic neurons, which were not the whole cells, in the whole mount DRGs. This caused ambiguity for quantification. In this revision, we performed RNAScope using DRG sections of the vehicle and DTA-treated A β TauDTR mice and quantified the tdTom+ neurons per DRG section instead (less ambiguity for quantifying Tdt+ DRG neurons). We found ~70% ablation of the tdTom+ neurons using this method, which were shown in the new Fig. 2B-D. To further understand which sub-population of LTMRs are specifically affected by this ablation strategy, as the reviewer suggested, we performed RNAScope *in situ* hybridization of *tdT*, *Ret*, *Calb*, and *Ntrk2*. We found a significant reduction of tdT+ neurons co-expressing A β RA LTMR markers (*Ret⁺* and

Calb1)⁺. There was a decrease of tdT⁺ neurons co-expressing *Ntrk2* too, but this change was not statistically significant. Thus, our study using Split^{Cre} genetic strategy mainly affects Ret⁺ A β RA LTMRs, which functions in modulating pain are novel and different from the *Ntrk2*⁺ ones that were previously published ⁴.

We have modified results in **page 11** and discussion in **page 28** to clarify this point.

-Nerve endings are reduced in number in plantar skin, but what about lanceolate endings in hairy skin?

Answer: The lanceolate endings are very complex in the wholemount hairy skin, and their exact numbers are hard to quantify. One LTMR gives rise to multiple lanceolate endings around different hair follicles, different types of LTMR innervate different types of hair follicles, and every hair follicle has some lanceolate endings. As a fact, very few published studies quantified the number of lanceolate endings due to these reasons. In addition, given the tiling⁵ and potential over branching of remaining LTMRs (one remaining neurons grow more lanceolate ending to cover the empty space) after ablation of some A β LTMRs in our experiments, this quantification won't provide meaningful readout regarding either cell types specificity or ablation efficiency. We believe that quantifying the number of Tdt⁺ DRG neurons and the central terminals are sufficient and more robust to show the overall ablation efficiency of our genetic strategy. We quantified mechanosensory end organs in the glabrous skin to show the local ablation efficiency, because our behavior assays focused on the hindpaw glabrous skin area.

- Given the mechanism by which DTA kills neurons, it is difficult to understand the conclusion the axon terminals are affected but the cell bodies are not.

Answer: DTA causes cell ablation by inhibiting protein translation. Since there are existing proteins in DRG neurons before DTA treatment, it takes time for DTR⁺ DRG neuron to be ablated by continuous DTA actions (1 week injection). In addition, inhibition of new protein synthesis would first cause retraction of neuronal terminals, hypotrophy of cell bodies, and eventually death of the entire neurons. Thus, it is very possible that for some DTA affected neurons, their cell bodies were still present, but they were hypotrophy and their axon terminals had retracted. **With the new quantification of tdT⁺ neurons per section, we observed more reduction (70%) (Fig. 2B-D). In addition, some of remaining tdT⁺ DRG neurons showed obvious reduction in soma shapes and sizes (hypotrophy, indicated by white arrows in Fig. 2C).**

We modified text in pages 11 and 12, Fig 2, and figure legend accordingly to indicate the new results.

(3) Fig. 3:

- The texture assay interpretation would benefit from comparing vehicle to DTA mice in each texture (rather than comparing textures within mouse groups).

Answer: Agreed. **We have changed the graph accordingly (new Fig. 3E) as suggested.**

- The von Frey data would be easier to view and interpret if the force (g) is plotted on a log scale.

Answer: We appreciate reviewer's suggestion. We re-analyzed our VFH data as suggested, and we didn't find any obvious difference in interpreting our results. Thus, we would like to stick with the conventional 50% PWT with force (g) as most publications used. Some examples using the log scale are shown here for comparison:

- Prior literature has shown that ablation of *Trkb*⁺ LTMRs that includes the Meissner endings (Neubarth et al Science 2020) eliminates mechanical allodynia (Peng et al Science 2017 and Dhandapani et al Nat Comms 2018). However, in this paper, since 50% of a subset of A-beta fibers are ablated, it is very difficult to interpret the behavioral data in the DTA mice due to the undefined population of cells being ablated.

Answer: Meissner's corpuscles are innervated by two distinct population of Aβ RA LTMRs, one Ret⁺ and one Trkb⁺³. In addition, most of the Trkb⁺ LTMRs are Aδ but not Aβ LTMRs^{1,2}. As discussed above, molecular, histological, and physiological characterization showed that this *Split*^{Cre} genetic strategy mainly targets the Ret⁺ Aβ RA LTMRs but not the Trkb⁺ Aβ or Aδ LTMRs. Thus, it is not surprising that our results are different from the previous publications. The functions of Ret⁺ Aβ RA LTMRs in mediating and modulating nociception are unknown and novel.

Our genetic ablation (~70% of DRG cell body and 80% of the fibers innervating dermal papillae) is efficient. In addition, the total number of Meissner's corpuscles is not changed in the ablated mice, but their innervation by tdT+ afferents decreased significantly (~62%) (Fig. 2H-J, and Supplementary Fig S1O-P). This efficient ablation of Aβ LTMRs is also supported by our behavior data. At baseline condition, our ablated mice displayed significantly reduced sensitivity to dynamic cotton swab and tape (Fig. 3C-D) and abnormal tactile discrimination (Fig. 1E), which clearly indicate that afferents for sensing gentle mechanical forces were affected. In contrast, the Dhandapani et al Nat Comms 2018 conducted adult ablation of TrkB+ afferents but found no baseline deficit in gentle mechanical force sensation. This is very surprising, given the knowledge that TrkB⁺ Aβ RA LTMRs innervate Meissner corpuscles and mediate gentle mechanical force sensation³. If looked at their figures carefully, their genetic tracing using TrkB-CreERT2 barely showed convincing Meissner corpuscle innervation. Thus, it seems that this previous publication but not our study may have an issue of efficient genetic labeling and ablation. The Neubarth et al Science 2020 study used developmental ablation of *Trkb*, which is known to be required for the formation of the entire Meissner corpuscles. Thus, for this study, all Meissner corpuscles were

gone, and both *Trkb*⁺ and *Ret*⁺ A β RA LTMRs were affected. This study didn't examine mechanical nociception in chronic pain model though. The *Peng et al Science 2017* paper studied microRNA functions in *TrkB*⁺ LTMRs. As discussed, *TrkB*⁺ LTMRs and *Split*^{Cre}-LTMRs are largely separated populations, and microRNAs could alter neuronal functions more than simple ablation or activation. In short, compared to these previous publications, our *split*^{Cre} genetic strategy preferentially targets adult *Ret*⁺ A β RA LTMRs as well as some A β SA LTMRs, which is different from previous studies and generates novel insights regarding functions of A β LTMRs in pain transmission.

In addition, the previous studies have not carefully compared the local vs global effects of A β LTMRs and taken this complexity into consideration, as we did in this study. In the study, we found that *split*^{Cre} positive A β LTMRs mediated mechanical allodynia, as shown by the specific optogenetic activation (Fig. 4). However, when these fibers were globally ablated, the “gating” effect of A β LTMRs became the dominant phenotype. Since other types of mechanosensory fibers, such as spared *TrkB*⁺ LTMRs, polymodal *MrgprD*⁺ fibers, and sensitized HTMRs (in chronic pain conditions) could still mediate sensation of gentle mechanical forces, the mechanical nociceptive responses and mechanical allodynia became stronger. We speculate that another reason that *TrkB*CreERT2 ablation (mainly affected A δ LTMRs) generated different phenotype is that A δ LTMRs may play a less important role in the “gate control of pain” compared to A β LTMRs.

We have modified the text, mainly discussion, in **page 28** of the revised manuscript to highlight the difference between our study and the previous ones.

(4) Fig. 4, the results from optogenetic stimulation of *Split-Cre* are entirely expected from von Frey stimulation.

Answer: Though different VFHs can deliver a range of mechanical forces, they can't isolate the function of one specific population of mechanosensory fibers. It becomes even more complicated after inflammation or nerve injury, in which high threshold mechanoreceptors (HTMRs) change molecular profiles ⁶ and display reduced mechanical threshold (peripheral sensitization). Instead, the optogenetic strategy allow us to activate the *Split*^{Cre} positive A β LTMRs specifically and solely before and after CFA treatment or nerve injury.

(5) Fig. 5:

- *There needs to be more neurons recorded to make a clear conclusion.*

Answer: As suggested, we have conducted additional optotagged-recording experiments and increased sample sizes for Figure 5. Please note that the occurrence of SA in control remains to be low, which is a feature of this genetic line, although we have increased total recordings to 22 recordings. We have incorporated the above ideas in the result section (pages 20-21).

- *The observation that, in chronic inflammation, rapidly adapting cells become slow adapting does not make physiological sense, because RA and SA properties are thought to be determined by static factors like genetic and end organ identity (i.e. Merkel/Ruffini endings are SA while Meissner/Pacinian endings are RA).*

Answer: It has long been known that RA responses of Pacinian corpuscles can become SA-like response following structure damage ⁷ (and PMID: 14194104). David Ginty and Bruce Bean have shown recently that the RA firing of afferent neurons of Meissner's endings is determined by Kv1.1 channels, and a decrease in the activity of these K⁺ channels convert RA firing to SA firing ⁸. Voltage-gated K⁺ channels are known to be down-regulated under pathological conditions including CFA-induced tissue inflammation ^{6,9}. Therefore, it is possible that RA responses of Meissner's endings change their phenotypical response from RA to SA-like responses following tissue inflammation induced by CFA. **We have incorporated the above ideas in the result section (pages 22).**

- Panel E has an inconsistency between the graph and the legend n numbers. Also, the line graphs should be made more visually distinct to distinguish between saline and CFA.

Answer: Changes have been done as suggested, and the error has been corrected.

(6) Fig. 6, the magnitude of response between B and F is the same post-optogenetic stimulation, suggesting that the stimulus is driving the effect regardless of the CFA treatment.

Answer: At baseline condition (without CFA, Fig. 6B), we tested effects of transient laser stimuli on mechanical threshold of *Split-ReachR2* mice using two different laser powers, 0.5 mW and 10 mW. These mice did not show obvious changes with 0.5 mW blue laser but did show a significant reduction of mechanical threshold with 10 mW blue laser. Thus, 0.5 mW laser power was chosen to probe mechanical changes after CFA treatment. In Fig. 6F, 0.5 mW blue laser was used to transiently stimulate the spinal cord dorsal column, and then mechanical thresholds of ipsilateral (CFA treated) or contralateral (control side) paws were tested in given intervals. Since the contralateral paws were not treated by CFA, their mechanical threshold after the 0.5 mW stimulation was unaffected, similar to the baseline condition. In contrast, in the CFA treated ipsilateral paw, where 50% paw withdrawal threshold was greatly reduced, transient 0.5 mW blue laser stimulation was able to significantly increase the mechanical threshold, indicating an antinociceptive effect. Thus, our results demonstrated that central stimuli of Split-ReaChR2+ afferents could alleviate pain. Given that our results from CFA treated and untreated paws were different, we respectfully disagree with the reviewer that the stimulus is driving the effect regardless of CFA treatment.

Reviewer #2:

Reviewer #2 (Remarks to the Author):

The role of large-diameter light-touch mechanoreceptor neurons in mechanical hyperalgesia remains unknown and controversial within the field. The Gate Control Theory, in which activation of large-diameter, light touch sensory neurons inhibits the transmission of small-diameter nociceptors within the spinal cord preventing pain sensation, was proposed over half a century to explain why light-touch alleviated pain and led to the development of clinical interventions such as Transcutaneous Electrical Nerve Stimulation. However, several studies

contradict this model and suggest that AB-LTMRs may mediate mechanical hyperalgesia of chronic pain. To reconcile these differences, Gautam and colleagues used refined genetic strategies coupled with electrophysiology, high-speed behavioral analysis, and optogenetics to examine the role of AB-LTMRs in the naïve and chronic injury state. Using an intersectional genetic strategy to label a large subset of AB-LTMRs while minimizing off-target recombination in non-neuronal glial cells (SplitCre; AdvilFlpO), Gautam, Yamada, Yamada et al found that genetic ablation of SplitCre-A β -LTMRs increased mechanical pain but did not affect thermosensation in acute and chronic inflammatory pain conditions, suggesting that A β -LTMRs play a modality-specific role in gating mechanical pain transmission. Conversely, optogenetic activation of these same neurons elicited higher pain scores following tissue inflammation in comparison to the naïve state, in which activation of the same subset led to behavioral responses with low pain scores. The authors suspected that the differing results of ablation vs activation of SplitCre-A β -LTMRs arises because of distinctive local and global roles of the SplitCre-A β -LTMRs in transmitting and alleviating mechanical hyperalgesia of chronic pain, respectively. In support of this model, the authors observed that broad activation of these same neurons activated at the level of the dorsal column alleviated mechanical hypersensitivity associated with chronic inflammation. Overall, the refined genetic approach taken in this study provides novel insights into the function of A β -LTMRs in chronic pain. Overall, this is a compelling study that will be of interest to researchers in the field. However, there are some concerns or suggestions discussed below that would help strengthen the study.

MAJOR CONCERNS:

(1a) *The study would benefit from further histological analysis of the SplitCre; AdvilFlpO line. Specifically, it seems that there is some amount of innervation by SplitCre neurons in layers I/II of the spinal cord (Figure 1F-G), although the authors state in the text that “sections showed that GFP+ central terminals did not innervate layers I/II”. Additionally, there are a handful of GFP+/NFH- terminals in the skin (Figure 1H) and an amount of double labeling with CGRP. It would be helpful for the authors to better characterize the GFP+/NF200- neurons in glabrous and hairy skin—are these CGRP+ HTMRs? Are any NFH- lanceolate endings (C-LTMRs or Ad-LTMRs) labeled by SplitCre?*

Answer: We thank reviewer 2 for raising this point. Please also see our answers to questions (1) of Reviewer #1 regarding further characterization of SplitCre;AdvilFlpO recombination patterns. **Briefly, we have performed multiplexed RNAScope *in situ* hybridization with additional markers of A β and A δ LTMRs (new Fig. 1A-D). We also performed IHC for CGRP⁺ fibers in plantar skin (Fig S1L, both low and high magnification images).** Though a few CGRP⁺ fibers innervate the dermal papillae around the Meissner’s corpuscles, they don’t really overlap with GFP+ fibers. From molecular characterization in the DRG cell bodies (Fig. 1 and supplementary Fig. 1), which is more straightforward and certain, majority of recombined neurons (>90%) are NF200+. The overlap between genetically traced fibers and NF200 was clear in supplementary Fig. 1N and 2J. In addition, **we have changed the color of NF200 staining to be red and showed high magnification pictures in new supplementary Fig. S1R-T** (also see answer to question (3)) to clarify this point. These molecular and histological results are consistent with random optotagged recordings, which showed 100% recorded fibers have conduction velocities in the A-beta range (Fig. 1L). Thus, even if *Split^{Cre}* recombines in some non-A-beta DRG neurons, their percentage must be very small (less than A β -SA-LTMRs in the glabrous skin, which is about 10%). For the central projections, **we changed the text to be “sections showed that few GFP+ central terminals**

innervate layers I/II”, as pointed out by the reviewer. We have also revised main text in pages 7-9, Fig. 1, Fig S1, and legends to incorporate the new results.

(1b) It would also be helpful to show GFP/NF200/Tuj1 staining of lanceolate endings in hairy skin to appreciate whether any other LTMR population is labeled.

We thank reviewer for this suggestion. We performed IHC for tdTom (Endogenous signal) plus Tuj1 (antibody) in the plantar skin of A β TauDTR mice (see below). In the hairy/back skin, we could only perform wholemount IHC for Tuj1 (see below) but not both, as our dsRed (For tdTom) and Tuj both were raised in Rabbit, and tdTom endogenous signals were not strong enough to be detected in the wholemount hair skin. Though nerve terminal staining and characterization are informative, the quantifications of different type are tricky. Given that our random opto-tagged recordings from both glabrous and hairy skin found 100% A beta fibers (Fig. 1L), and since the immunostaining of NF200 found in more than 90% EFYP DRG neurons, the chance of recombination in other cutaneous non-A-beta types of LTMRs is low. In addition, given that our behavior assays focused on the paw glabrous skin, it makes more sense that we focused on nerve terminal characterization in that skin area.

(1c) *It is difficult to really appreciate the overlap between NFH and GFP in lanceolate endings from the image shown in S1G.*

We agree with the reviewer that it is difficult to appreciate the overlap between NFH and GFP in lanceolate endings from the images shown in the old S1G, because they were shown in blue and green. **In the revised supplementary figure 1, we changed the colors to be red (NF200) and green (GFP) and representative lanceolate images at higher magnifications shown in figures S1R-T.** These changes would make it easier to appreciate the overlap.

(1d) *Understanding whether CGRP+ neurons in the skin are labeled by SplitCre would be helpful in interpreting why optogenetic activation of this population (at 10 and 20 mW) leads to nocifensive responses in the naïve state. Also for Fig 1A, it would be helpful to show CGRP staining in red so that it would be easier to appreciate any potential overlap between GFP/CGRP.*

We thank the reviewer for this feedback. **We changed CGRP staining in red, as suggested, in the revised Fig. S1E and S1M, and showed high magnification image of skin innervation (Fig. S1M).** Though CGRP+ fibers were found the dermal papillae region near Meissner's corpuscles, they didn't really overlap with GFP+ Meissner corpuscle afferents.

We found a low pain score (less than 1) when using 10 mW and 20 mW blue laser to activate this population at the baseline condition (Fig. 4D). We didn't use these light intensities for further experiments because 5 mW light gave a cleaner baseline, but it didn't mean that such a low "pain score" necessarily indicated "nocifensive response". Please note that we recorded and quantified mouse paw withdrawal reflexes using high-speed imaging (500 to 1000 frames per second). With regular videotaping cameras as most labs do, these responses that we caught would be negligible. **In the revised manuscript, we added baseline behavior results of *TrpV1-Cre;Advilin-FlpO;ReaChR (TrpV1-ReaChR)* mice in response to 5 mW blue laser as a positive control for "nocifensive responses" (orange bars in Fig. 4B-D and Fig. 4H-I).** As we can see, 5 mW blue laser stimuli to *TrpV1-ReaChR* mice triggered much obvious responses, with significantly shorter latency (~200 ms) and higher pain score (averaged ~3.5 (Fig. 4C-D)). Thus, the 10 mW and 20 mW blue laser triggered responses at the baseline condition are very minor.

We have revised main text in pages 8, 18-19, Fig. S1, Fig. 4, and legends to incorporate the new results.

(2) *The authors claim that "in chronic inflammation, the electrophysiological properties of many RA AB-LTMRs became SA-like, which could be one of the underlying mechanisms for triggering stronger local activity and inducing mechanical hyperalgesia in the CFA model", does not seem supported by the data for the following reasons:*

1. Following CFA, the number of impulses for AB RA-LTMRs increased from 1 to 2— this is far from the value reached by AB SA-LTMRs.

2. *The reduction in RA and increase in SA responses following CFA may be due to the decrease in GFP+ fibers innervating Meissner corpuscles, which may render these neurons unresponsive to mechanical stimulation. As such, there may just be fewer AB RA-LTMRs following CFA.*

3. *Along these lines, the increase the authors describe for the SA neurons is based on a recording of 1 SA neuron in the control condition. I think the potential increase in SA firing is interesting and therefore it would be worth trying to recording from 2-3 more SA neurons in the control condition to enable statistical comparisons.*

Answer:

1. Following CFA, the number of impulses for Abeta RA-LTMRs increased from 1 to 2 in the dynamic phase of mechanical stimulation. These impulses in Figure 5B were still classified as RA-LTMRs but not SA-LTMRs. SA-LTMRs were classified when impulses occurred in both dynamic and static phases of mechanical stimulation. **We added a sentence to clarify this point in the method in page 44.**

2. We appreciate that the reviewer provided us with an alternative interpretation for our data (Figure 5E) in terms of the changes in the proportion of RA vs SA. In the control group, mechanically evoked responses were mostly RA type and very occasionally we observed SA in the glabrous skin (3/22). In contrast, in the CFA group, we encountered more SA type responses (6/11) and less RA type responses (5/11), but not just the reduction of RA. Therefore, a more reasonable interpretation for our data is that some RA LTMRs becomes SA-like fibers.

In the previous version, we only showed 1 SA fiber in the control group because of its very low occurrence as a feature of this *Split^{Cre}* line. As requested by the reviewer 1, **we have performed new experiments and increased total numbers of SA fibers (n = 3) and total control group of 22 fibers (19 RA vs 3 SA)**.

(3) *The authors show activation of the dorsal column alleviates pain (paw withdrawal threshold, Figure 6F) but this behavioral effect lasts 5 to 30 minutes after optogenetic stimulation. The authors should comment on what they think this is happening for this prolonged period of time.*

Answer: We conducted 15 minutes optogenetic stimulation of the spinal cord and found an effect lasting 5 to 30 minutes after the light stimuli. We think that this lasting effect could be due to the synaptic potentiation (PMID: 24162508, 36232917). **We added a sentence in the result section to discuss this point in page 25, as the reviewer suggested.**

(4) *More statistical detail is needed (the ANOVA tests should report the p value, F statistic, and degrees of freedom in addition to the post-hoc test p value)- regarding the test in 4I would be nice. Was an ANOVA done here? If so, why do the authors think the littermate CFA animals are not different from the CFA animals with ReaChR activation in this conditioned place assay? Do the CFA animals at baseline prefer one side of the chamber?*

Answer: We apologize for the confusion here. For this set of experiments, littermate control mice were those, in which one genetic element, *Split^{Cre}*, *Advil^{FlpO}*, or *ReaChR* was absent. In another

word, the littermate control mice didn't express ReaChR and could not be activated by light stimuli. In addition, we did not pair the CFA-administration with the specific chamber, but the blue light stimuli with a specific chamber. The littermate control group with CFA simply served as a negative control, in which mice felt spontaneous pain caused by CFA-induced inflammation but not the blue light.

We appreciated reviewer's comments about statistical analysis with Figure 4I. As suggested, in this revision, we have conducted statistical comparison between blue light stimulated CFA treated *Split-ReaChR2* mice vs blue light stimulated CFA treated control mice or blue light stimulated saline treated *Split-ReaChR2* mice using Two-way ANOVA followed by Tukey's multiple comparisons test. We found a significant difference in place preference for the experimental group after stimulation compared to the two negative control groups ($p=0.004$ or 0.029 , Fig. 4I). In addition, we have also added CPP results of untreated TrpV1-ReaChR2 mice ($n=4$), which showed a significant avoidance to the blue laser stimuli (Figure 4I, orange bar), as the positive control group.

We have modified text in the result section in **page 19**, Figure 4, and legends, to clarify the "littermate control mice" and to incorporate the new results.

Minor:

-The description of the genetic strategy in lines 194-196 is confusing and should be clarified. Currently the way it is written makes it sound like Tau-DTR is just Cre-dependent.

Answer: Thanks for this feedback. In this revision, we have changed the text in **page 10** to make it clear: we crossed *Split^{Cre};Advil^{FlpO}* double mice to homozygous reporter mice, which contained a Cre- and FlpO- double-dependent DTR allele (*Tauds-DTR^{ff}*) (the human diphtheria toxin receptor (DTR) driven by a pan neuronal Tau promoter) (Duan et al., 2014) and a Cre-dependent Ai9 (*Rosa-tdTomato^{ff}*) reporter allele. The resulting quadruple mice expressed DTR in neurons co-expressing *Split^{Cre}* and *Advil^{FlpO}*, while tdTomato expression indicated the recombination activity of *Split^{Cre}*.

-The results of the texture preference assay (Fig. 3E) suggest the DTR-treated mice have a heightened sensitivity to floor texture, in comparison to the control animals. This is the opposite of the authors interpretation (265-267). Could the author please elaborate on the discrepancy in light touch sensitivity between the gentle touch assays and this texture preference assay?

Answer: Vehicle-treated mice showed no significant preference to the two floor textures, the loop and hook surfaces (Fig. S3E), used for this experiment. In contrast, the ablated mice (ablation of *Split^{Cre}* A-beta LTMRs) showed increased preference towards the chamber with loop surface. Loop texture is softer and smoother than the rough-hook floor, when we examined under the microscope (Fig. S3E a-b). Our interpretation is that with a reduction of inhibition from *Split^{Cre}* A-beta-LTMRs ("gate control theory of pain"), the ablated mice had enhanced nociception from the rough floor texture and thus tended to avoid this chamber. We have switched the graph bar orders in this revision (new Fig. 3E), comparing the Vehicle vs. DTA groups side by side for each

chamber (Please also see the response to question #3 of Reviewer #1). We also added a sentence in the result section to explain this point (page 14).

-It would be helpful for the authors to compare the pain score of SplitCre activation in the naïve and chronic pain state to optogenetic activation of a nociceptive population. Is a pain score below 1 painful? It is unclear why optogenetic activation of SplitCre neurons with higher intensity light would lead to nocifensive behaviors (Fig 4D). Could the authors comment on this in the paper?

Answer: This is a great point. As mentioned above, we have added behavior results of TrpV1-ReaChR mice in response to 5 mW blue laser stimuli as the positive nociceptive control (Fig. 5B-D and 5I). Please also see our response to the major point 4 above.

-The legend for Figure 4I describes more than the one figure panel and should be edited.

Answer: Thanks for pointing this out. We made correction in the revision.

-It is challenging to appreciate the location of the Merkel cell/SAI complex and the Meissner corpuscles in the confocal images in S5D-G either because of low magnification or unideal sectioning plane. Perhaps these examples can be swapped with more representative examples that clearly show examples of Merkel cell complex and Meissner corpuscles.

Answer: Thanks for pointing this out. We have added images with better section planes and also inset images with higher magnifications to demonstrate the point (supplementary Fig. 5D-G).

-In Figure 5E the Saline SA category says there were 2 neurons recorded but in the text and in 5D and 5G, the authors describe only 1.

Answer: We have performed additional recordings with baseline condition for this revision, and now the total numbers of SA type are 3 as shown in Fig. 5D, E, and G. The new total numbers of RA type in control group are 19 in Fig. 5E&F. Changes have been made in the related figures and the text.

Reviewer #3:

This is a well conducted study by Luo's group in collaboration with Gu's group. The role of low-threshold A β -low threshold mechanoreceptors (LTMRs) in pain modulation under the physiological and pathological conditions are not fully understood. In this study, authors combined intersectional mouse genetics, opto-tagged electrophysiology recordings, and high-speed imaging behavioral assays to investigate the contribution of A β -LTMRs to pain transduction and transmission. They showed both global ablation and local activation of A β -LTMRs promoted mechanical hyperalgesia. They also showed that global activation of these

afferent fibers via dorsal column activation alleviated inflammatory pain via activation of inhibitory neurons in the spinal cord dorsal horn. It is very useful to generate triple heterozygous SplitCre;AdvilFlpO;RosaReaChRf/+ (SplitCre -A β ReaChR) mice as a research tool. The electrophysiological characterization is also very impressive. It is interesting to show that in chronic inflammation the electrophysiological properties of many RA A β -LTMRs became SA-like. Overall, the findings are interesting and provide new mechanistic insights into A β -LTMR mediated pain control.

Major:

1. Fig. 1A and Fig. 2B. There is some inconsistency. It seems many more neurons are labeled in Fig. 2B. than in Fig. 1A.

Answer: The previous Fig. 1A showed IHC using a 20-micron thick DRG section, while the previous Fig. 2B showed an image of a whole mount DRG, which had a diameter ~600 microns and contained samples of ~30 sections. Thus, Fig. 2B image had far more tdT+ neurons as anticipated from the different methods. **In this revision, we replaced whole-mount DRG images (old Fig. 2B) with RNAscope images using 20-micron thickness DRG section (new Fig. 2B), which nicely demonstrated the ablation efficiency. In addition, we put the new RNASCOPE multiplex images, as suggested by the reviewer 1, in the Fig. 1A-C, and moved the IHC images to the Fig. S1E-H).**

2. Line 213: only a 45.3% decrease of tdTomato+ DRG neurons was found after DTA treatment. Do you think the expression pattern of your marker may change after inflammation or nerve injury?

Answer: Previously, we used whole mount DRGs to quantify number of tdT+ DRG neurons. However, there were tdTom signals generated by apoptotic neurons but not the real whole cells, which caused ambiguity for quantification. In this revision, **we performed RNAscope using DRG sections of the vehicle and DTA-treated A β TauDTR mice and quantified the tdTom+ neurons per DRG section instead (less ambiguity for quantifying Tdt+ DRG neurons). We found ~70% ablation of the tdTom+ neurons using this method, which were shown in the new Fig. 2B-D.** Please also see our answers to question 2 of the reviewer 1. Several studies have found changes in the gene expression of DRG neurons after nerve injury (PMID: 32810432). Nevertheless, we feel that expression of our marker, genetic labeling of ReaChR, may not be dramatically changed, because we didn't see an overall changes of peripheral nerve terminal morphologies in the glabrous skin sections (for an example, no increase of Merkel cell innervating GFP+ fibers, no obvious GFP+ free nerve terminals in the epidermis after CFA treatment, Fig. S5) and by optotagged physiological recordings (still A-beta LTMR fibers, despite a change of relative abundance of SA and RA units, Fig. 5).

3. It appears the ablation was only conducted in naïve animals. It is also important to know if A β -LTMR ablation would also affect mechanical pain after CFA or nerve injury.

Answer: Agreed. We showed the effect of ablation on mechanical hyperalgesia in CFA-induced inflammation (Fig. 3G). **In this revision, we also added new data with nerve injury, a medial plantar nerve ligation (MPNL) neuropathic pain model¹⁰ in the supplementary Fig. 3E.** Please note that the mechanical threshold of mouse paw was so low (close to 0) 3 days after the nerve

ligation and didn't recover during the experimental period time (supplementary Fig. 3E, the "floor effect"), the ablation of SplitCre positive A-beta LTMRs did not render further significant changes.

The new MPNL result was described in **page 15** of the revised manuscript.

4. Figure 4A-F: It appears the control group in animals with CFA or nerve injury (without stimulation or with sham stimulation) is missing.

Answer: We thank reviewer for raising this point. **1)** For optogenetic experiments using ChR2, people usually use green light as the sham light for control. However, ReaChR is red shifted and has a much broad activation spectrum¹¹, from blue light (the Luo lab used for behavior assays in this study) to orange/red light (the Gu lab used for physiological recording in this study). Green light (falls in between the blue and red) or all other commonly available laser or LED lights can activate ReaChR, so we could not do the sham light control experiments. **2)** The "no light control" has been included in our experimental protocol. This is what we had in the method section (page 68): Mice were acclimated on an elevated perforated wire-mesh platform daily for 1 h in transparent rectangular Plexiglas chambers either for 2 (Plantar stimulation) or 3 days (after shaving the nape of the neck and back). On the day of the experiment, they were habituated for 45 min before peripheral 473 nm blue laser (Shanghai Laser and Optics Century, BL473T8-150FC/ADR-800A) stimulation. When mouse was still, a blue laser (5-20 mW, 10 Hz square wave, waveform generator (hp Hewlett Packard 15MHZ Function Waveform Generator, 33210A)), was shined upon the plantar surface below the wire-mesh space (the distance between paw and the laser cord-outlet tip is ~2 mm). Thus, **the control phase (without light stimulation) of all mice were included as an intrinsic part of our protocol.** Since we only conducted ad quantified behavior of fully habituated mice, which were still and showed no movement in this baseline phase, data from this control were 0. **3)** It is not feasible to have a separate group of mice as "no light control" either. Without light stimuli, which were used as a time reference for analyzing optogenetic behavior assays, there will be no rational for behaviors in which period of time to be picked as the control.

We added a sentence in this method section (**page 68**) to make this point clear.

5. Figure 6D: Is c-Fos expression shown from the lateral or medial spinal cord? A low magnification followed by high magnification will be helpful.

Answer: Please see the Suppl. Fig S6M-P for the low-magnification images. The magnified images were from medial side, as shown by inset.

Other comments

-I am curious if you also examined mechanical itch? Aβ-LTMRs have been implicated in regulating mechanical itch (PMID: 31324538).

Answer: Thank you for your interesting question. Yes, we did the experiments but didn't include data in this manuscript, as they were too variable to draw a clear conclusion. We administered 6 filaments of various forces (0.008 to 1 mN) 5 times at neck nape as well as the posterior ear of habituated mice and recorded using high-speed imaging videos (1000 fps). There was a lot of variability in the scratching and other related behaviors (wiping, head shake). Below the data (Blue bars: Vehicle-treated; Red bars: DTA-treated $A\beta$ *TauDTR* mice):

-Line 303: The ablated mice displayed significantly increased scratching behavior in response to the tape. Scratching may be related to itch?

Answer: That is possible. We speculate that due to the loss of $A\beta$ LTMR inhibition, activation of other unaffected LTMRs in the hairy skin may lead to an enhanced itch/scratching-related behaviors. We didn't comment this in the text, because it is only a speculation.

-Line 350: same stimuli evoked stronger responses of SplitCre-A β -LTMRs in CFA-induced chronic inflammatory pain condition. I wonder if this increase also occurs in the contralateral side. Bilateral mechanical allodynia was reported in the CFA model.

Answer: At our hand, we didn't notice any obvious nocifensive behavior when unaffected/contralateral hindpaws were stimulated using 5 mW blue laser (n=5, data not shown). It is possible that higher intensity of light might needed to trigger hypersensitivity of contralateral paws for this situation. Alternatively, the contralateral mechanical allodynia may be mediated through other types of sensory afferents.

References

- 1 Sharma, N. *et al.* The emergence of transcriptional identity in somatosensory neurons. *Nature* **577**, 392-398 (2020). <https://doi.org:10.1038/s41586-019-1900-1>
- 2 Rutlin, M. *et al.* The cellular and molecular basis of direction selectivity of Adelta-LTMRs. *Cell* **159**, 1640-1651 (2014). <https://doi.org:10.1016/j.cell.2014.11.038>
- 3 Neubarth, N. L. *et al.* Meissner corpuscles and their spatially intermingled afferents underlie gentle touch perception. *Science* **368** (2020). <https://doi.org:10.1126/science.abb2751>
- 4 Dhandapani, R. *et al.* Control of mechanical pain hypersensitivity in mice through ligand-targeted photoablation of TrkB-positive sensory neurons. *Nat Commun* **9**, 1640 (2018). <https://doi.org:10.1038/s41467-018-04049-3>
- 5 Kuehn, E. D., Meltzer, S., Abaira, V. E., Ho, C. Y. & Ginty, D. D. Tiling and somatotopic alignment of mammalian low-threshold mechanoreceptors. *Proc Natl Acad Sci U S A* **116**, 9168-9177 (2019). <https://doi.org:10.1073/pnas.1901378116>
- 6 Renthal, W. *et al.* Transcriptional Reprogramming of Distinct Peripheral Sensory Neuron Subtypes after Axonal Injury. *Neuron* **108**, 128-144 e129 (2020). <https://doi.org:10.1016/j.neuron.2020.07.026>
- 7 Gray, J. A. & Matthews, P. B. A comparison of the adaptation of the Pacinian corpuscle with the accommodation of its own axon. *J Physiol* **114**, 454-464 (1951). <https://doi.org:10.1113/jphysiol.1951.sp004636>
- 8 Zheng, Y. *et al.* Deep Sequencing of Somatosensory Neurons Reveals Molecular Determinants of Intrinsic Physiological Properties. *Neuron* **103**, 598-616 e597 (2019). <https://doi.org:10.1016/j.neuron.2019.05.039>
- 9 Biet, M., Dansereau, M. A., Sarret, P. & Dumaine, R. The neuronal potassium current I(A) is a potential target for pain during chronic inflammation. *Physiol Rep* **9**, e14975 (2021). <https://doi.org:10.14814/phy2.14975>
- 10 Sant'Anna, M. B. *et al.* Medial plantar nerve ligation as a novel model of neuropathic pain in mice: pharmacological and molecular characterization. *Sci Rep* **6**, 26955 (2016). <https://doi.org:10.1038/srep26955>
- 11 Lin, J. Y., Knutsen, P. M., Muller, A., Kleinfeld, D. & Tsien, R. Y. ReaChR: a red-shifted variant of channelrhodopsin enables deep transcranial optogenetic excitation. *Nat Neurosci* **16**, 1499-1508 (2013). <https://doi.org:10.1038/nn.3502>

REVIEWERS' COMMENTS

Reviewer #1 (Remarks to the Author):

The authors have revised the work, included new detail about the transgenic approach and it's specificity. There are new data that help clarify many of the issues that were raised. I still remain skeptical that an RA neuron transforms to and SA (Reviewer #2 raises a similar question). However, this can wait for further studies and is no reason to hold things up further. I have no additional suggestions.

Reviewer #3 (Remarks to the Author):

This revision has been greatly improved, and the authors have included many panels of new data for additional characterization of their transgenic mice. They also included behavioral data in a neuropathic pain model.